# Thermal Stress Formation in a Functionally Graded Al₂O₃-Adhesive Single Lap Joint Subjected to a Uniform Temperature Field

Mustafa Kemal Apalak [1,*] and Junuthula N. Reddy [2]

1   Department of Mechanical Engineering, Erciyes University, 38280 Kayseri, Türkiye
2   Department of Mechanical Engineering, Texas A&M University, College Station, TX 77843-3123, USA; jnreddy@tamu.edu
*   Correspondence: apalakmk@erciyes.edu.tr

**Abstract:** This study investigates the strain and stress states in an aluminum single lap joint bonded with a functionally graded Al₂O₃ micro particle reinforced adhesive layer subjected to a uniform temperature field. Navier equations of elasticity theory were designated by considering the spatial derivatives of Lamé constants and the coefficient of thermal expansion for local material composition. The set of partial differential equations and mechanical boundary conditions for a two-dimensional model was reduced to a set of linear equations by means of the central finite difference approximation at each grid point of a discretized joint. The through-thickness Al₂O₃-adhesive composition was tailored by the functional grading concept, and the mechanical and thermal properties of local adhesive composition were predicted by Mori–Tanaka's homogenization approach. The adherend–adhesive interfaces exhibited sharp discontinuous thermal stresses, whereas the discontinuous nature of thermal strains along bi-material interfaces can be moderated by the gradient power index, which controls the through-thickness variation of particle amount in the local adhesive composition. The free edges of the adhesive layer were also critical due to the occurrence of high normal and shear strains and stresses. The gradient power index can influence the distribution and levels of strain and stress components only for a sufficiently high volume fraction of particles. The grading direction of particles in the adhesive layer was not influential because the temperature field is uniform; namely, it can only upturn the low and high strain and stress regions so that the neat adhesive–adherend interface and the particle-rich adhesive–adherend interface can be relocated.

**Keywords:** functionally graded material; Al₂O₃; adhesive; thermal stress; elasticity theory; finite difference method; micro particles

## 1. Introduction

The adhesive bonding technique has been used successfully to join similar and different materials. In general, adhesive joints are designed so that they can withstand static and dynamic loadings [1,2]. However, today's adhesives can serve at cryogenic, low, and high temperatures. The thermal loads result in non-uniform thermal stress distributions, which appear in a discontinuous manner in vicinities of adhesive–adherend interfaces due to incompatible thermal strains as a result of different mechanical and thermal properties of the materials on both sides of bi-material interfaces [3]. A uniform or non-uniform temperature distribution, a non-uniform material property distribution, chemical and physical changes induced in the adhesive material during the adhesive curing process, the expansion of adhesive with changes in moisture and temperature levels also result in thermal stresses in adhesive joints [4,5].

In various types of mechanical loadings, the adhesive joints undergo stress concentrations, called edge effects, around the free edges of adhesive–adherend interfaces while the normal and shear stresses remain uniform at low levels in a large overlap region and

increase uniformly towards the free edges of the adhesive layer and then become peak near these free edges. A large overlap region can also experience high stresses except for the adhesive-free edges depending on the type of thermal loading. In order to reduce the stress concentrations and to improve the joint strength, some joint geometry-specific measures were considered by adjusting the stiffness of adherends around these critical regions. However, the proposed geometrical measures to relieve these peak stresses can cause losses in the overall stiffness and strength of adhesive joint [1,2,6].

A layered composite structure, which can be joined easily by adhesive bonding technique, can exhibit better thermal and mechanical properties to single-composite material. Nevertheless, a thermal load can result in critical stress concentrations occurring along bi-material interfaces due to the sharp discontinuities in the material properties [7,8]. The concept of functionally graded materials (FGMs) is already utilized by biological interfaces in nature in order to reduce stress concentrations along bi-material interfaces. FGMs aim to achieve an equivalent performance to that of single-phase materials by unifying the better properties of the constituent phases with one- or more-dimensional continuously varying material composition; consequently, this can remove sharp discontinuity along the bi-material interface and relieve sudden jumps in the thermal stresses along bi-material interfaces [9–11].

Although the concept of FGMs is new, a large number of research studies have been carried out, and this field continues to expand fast [12,13]. Today, this concept can also be implemented to reduce stress concentrations appearing along the adherend–adhesive interfaces of the adhesive joints serving under static, dynamic, and thermal loads, i.e., use of functionally graded adherends and adhesives [14]. The stress distribution and peak stress levels can be controlled by tailoring one- or two-dimensional composition variation of adherends as well as an adhesive with one or more other constituents [14–16]. This method is especially helpful for relieving thermal stresses due to thermal loads.

Mathematical models and solutions on the thermal residual stress analysis of adhesive joints with functionally graded adherends have been continuously improved, and the functionally graded adherends were reported to relieve both stress and strain distributions and levels in the adhesive layer as well as in the adherends even though the adhesive layer was still in a functionally ungraded state [17].

An adhesive layer with variable modulus, which requires at least the use of two adhesives with different mechanical and thermal properties, has been proposed to relieve high-stress concentrations at the free edges of the adhesive layer and to have more uniform stress distributions along the overlap region [18–23]. This can be considered as a stepped functionally graded adhesive layer; namely, a stiff adhesive in the middle portion of the overlap region and a flexible adhesive around the free edges of the overlap region are applied. The concept of using multi-modulus adhesives can improve the overall joint strength and can also be implemented to the thermal stress problems of the adhesive joints to operate at low and high temperatures [24]. Namely, a high-temperature adhesive in the middle of the overlap region keeps the strength by transferring the entire load, whereas a low-temperature adhesive withstands loads at low temperatures by causing the high-temperature adhesive to undergo moderate stress levels [18,25,26].

The dual or mixed adhesive technique also brings some drawbacks. The stiff adhesive may tend to displace the ductile adhesive under the applied pressure; therefore, the bonded joint may be worse off than using the ductile adhesive alone in the manufacturing stage [18]. Another common method is to add various reinforcements of different scales, which are harder, stiffer, and more strength than adhesive material, to the adhesive layer. Therefore, the mechanical properties, electrical and thermal conductivities of adhesive material can be improved suitably depending on its application area [27]. In general various fillers at a specific weight/volume fraction are distributed uniformly through adhesive material, and a homogeneous distribution of fillers is desired as possible. The stress analyses of this reinforced adhesive layer under mechanical and thermal loadings are performed using

its mechanical and thermal properties, which are predicted by various experimental or continuum mechanics-based homogenization methods [28–31].

New theoretical analyses propose the use of a continuous adhesive grading, such as for modulus or coefficient of thermal expansion, along one or two coordinate directions of the adhesive material, and indicate that the improved strength of adhesive joints can be achieved by controlling stress concentrations with an existing optimum material grading rule [14,15]. Nevertheless, the material grading distribution rules are not practical right now for production purposes. The implementation of fused deposition modeling (3D printing) to the adhesive joints is now promising for the production of a functionally graded adhesive layer [32–35]. Consequently, the practice of functionally graded adhesive is still in the development stage and needs many theoretical and experimental studies that consider all aspects of adhesive material and adhesive joint.

In this study, the strain and stress states in an aluminum single lap joint bonded with a through-thickness functionally graded $Al_2O_3$ micro particle reinforced adhesive layer were investigated under a uniform temperature field (Figure 1). The spatial derivatives of Lamé constants and the coefficient of thermal expansion of local adhesive composition were considered in Navier equations of two-dimensional elasticity theory. The mechanical and thermal properties of the local adhesive composition were predicted by Mori–Tanaka's homogenization approach. The set of partial differential equations was solved with mechanical boundary conditions at each grid point of a discretized joint using the central finite difference approach. The effect of gradient power index controlling the through-thickness volume-fraction variation of particles in the local adhesive composition was also analyzed on the strain and stress states of the adhesive layer and adherends.

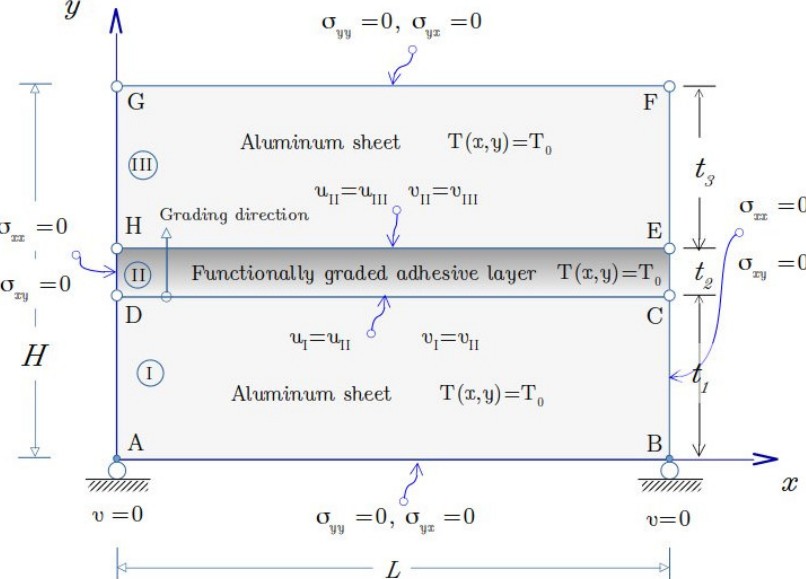

**Figure 1.** Boundary conditions of aluminum single-lap joint bonded with a through-thickness functionally graded adhesive layer.

## 2. Local Material Properties

The local composition of a functionally graded adhesive can be tailored along one or more spatial coordinate directions by means of a power function [9]. A continuous variation for any local mechanical property can be achieved; consequently, this can relieve discontinuities in thermal stresses which are encountered along the interfaces of layered material structures due to mismatches of mechanical and thermal properties.

The local mixture of both adhesive and micro-size ($Al_2O_3$) powders at any position through the adhesive thickness can be defined in terms of the volume fraction of ($Al_2O_3$) particles mixed through a two-component epoxy-based adhesive between a particle-rich

adhesive interface and a neat adhesive interface. A well-known mixture rule can be implemented for the local composition of a particle-reinforced adhesive as

$$V_a(y) + V_p(y) = 1, \tag{1}$$

where $a$, $p$, $V_a(y)$ and $V_p(y)$ indicate adhesive and (Al$_2$O$_3$) particles and their volume fractions through the adhesive thickness. The maximum volume fraction of particles $V_{max}$ in the vicinity of the particle-rich–adhesive interface is limited to a reasonable range of 0.01 and 0.1 because the adhesion between adhesive and aluminum adherend can not deteriorate. The volume fraction of (Al$_2$O$_3$) particles at any $y$—position through the adhesive thickness can be defined as

$$V_p(y) = \left(\frac{\bar{y}}{t_2}\right)^n V_{max} \quad \text{or} \quad V_p(y) = \left(1.0 - \left(\frac{\bar{y}}{t_2}\right)^n\right) V_{max}, \tag{2}$$

where $t_2$ is the adhesive thickness, $\bar{y} = y - t_2$ is the position relative to the lower adhesive interface. The power index $n$ provides a desired variation form of the volume fraction of particles through the adhesive thickness in a range of 0.1 and 12.0. As shown in Figure 2, the first equation provides a composition variation from a neat lower adhesive interface to a particle-rich upper adhesive interface, and the second one acts in the reverse sense.

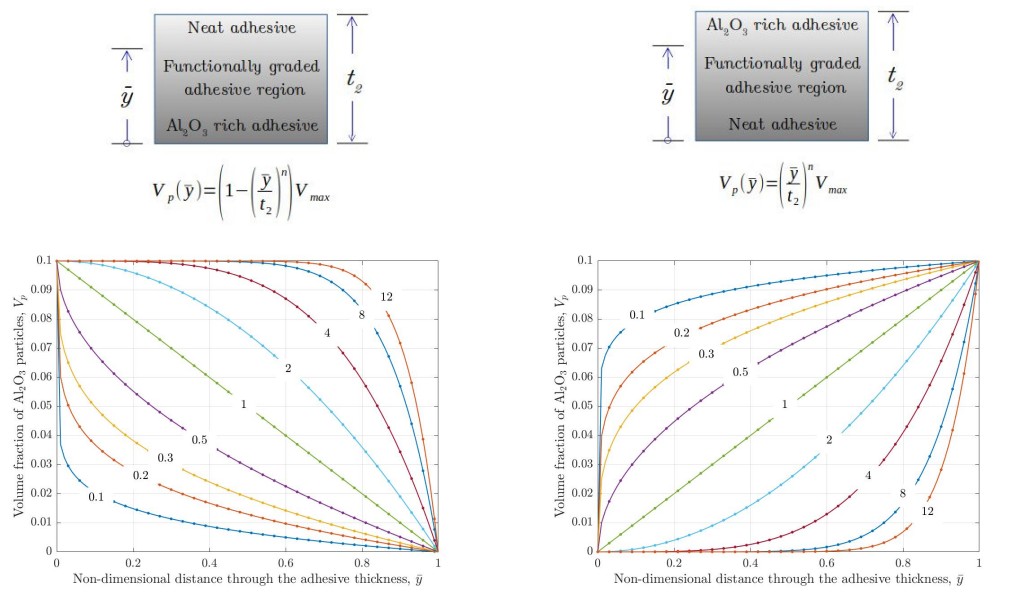

(**a**) Al$_2$O$_3$ rich adhesive (PRA) $\rightarrow$ neat adhesive (NA)    (**b**) Neat adhesive (NA) $\rightarrow$ Al$_2$O$_3$ rich adhesive (PRA)

**Figure 2.** The volume fraction $V_p(\bar{y})$ variations of Al$_2$O$_3$ particles through the thickness of a functionally graded adhesive layer for different gradient power index ($n$) values ($V_{max} = 0.1$).

The mechanical and thermal properties of the local adhesive and particle mixture can be predicted using the Mori–Tanaka homogenization approach [28,29]. Thus, at any position $\bar{y}$ through the adhesive thickness, the overall bulk modulus

$$K(\bar{y}) = K_a + \frac{V_p(K_p - K_a)}{1 + (1 - V_p)\frac{3(K_p - K_a)}{3K_a + 4G_a}}, \tag{3}$$

the shear modulus

$$G(\bar{y}) = G_a + \frac{V_p(G_p - G_a)}{1 + (1 - V_p)\frac{G_p - G_a}{G_a + f_1}}, \tag{4}$$

where

$$f_1 = \frac{G_p(9K_p + 8G_p)}{6(K_p + 2G_p)}, \tag{5}$$

the modulus of elasticity

$$E(\bar{y}) = \frac{9KG}{3K + G}, \tag{6}$$

Poisson's ratio

$$\nu(\bar{y}) = \frac{3K - 2G}{2(3K + G)} \tag{7}$$

For isotropic particle-reinforced composite materials Wakashima-Tsukamoto [30] and Levin [31] propose the overall coefficient of thermal expansion in terms of the overall bulk modulus as

$$\alpha(\bar{y}) = \alpha_a + \left(\frac{1}{K} - \frac{1}{K_a}\right) \frac{\alpha_p - \alpha_a}{\frac{1}{K_p} - \frac{1}{K_a}} \tag{8}$$

## 3. Mathematical Model and Solution Method

Let $x_1$ and $x_2$ be spatial coordinate variables. In order to solve an elasticity problem, we need to consider the fundamental equations of elasticity theory [36] as follows:

Equilibrium

$$\frac{\partial \sigma_{ji}}{\partial x_i} + F_i = 0, \tag{9}$$

Strain-displacement relations

$$\varepsilon_{ij} = \frac{1}{2}\left(\frac{\partial u_i}{\partial x_j} + \frac{\partial u_j}{\partial x_i}\right), \tag{10}$$

Stress-strain relations

$$\sigma_{ij} = 2\mu\varepsilon_{ij} + \lambda\delta_{ij}\varepsilon_{nn} - \delta_{ij}(3\lambda + 2\mu)\alpha\bar{T}, \tag{11}$$

where $\delta_{ij}$ is kronecker delta and Lamé's constants

$$\lambda(x_2) = \frac{\nu E}{(1 + \nu)(1 - 2\nu)}, \tag{12}$$

$$\mu(x_2) = \frac{E}{2(1 + \nu)}, \tag{13}$$

the coefficient of thermal expansion $\alpha = \alpha(x_2)$, Poisson's ratio $\nu = \nu(x_2)$, the modulus of elasticity $E = E(x_2)$, the temperature difference $\bar{T} = T_o - T_{ref} = $ constant and the volumetric strain

$$\varepsilon_v = \varepsilon_{nn} = \frac{\partial u_k}{\partial x_k} = \varepsilon_{11} + \varepsilon_{22} + \varepsilon_{33} \tag{14}$$

Substituting Equations (10) and (14) into Equation (11) yields

$$\sigma_{ij} = \mu\left(\frac{\partial u_i}{\partial x_j} + \frac{\partial u_j}{\partial x_i}\right) + \lambda\delta_{ij}\frac{\partial u_k}{\partial x_k} - \delta_{ij}(3\lambda + 2\mu)\alpha\bar{T} \tag{15}$$

Finally, the substitution of Equation (15) into Equation (9) for $(F_i = 0)$ gives

$$\frac{\partial}{\partial x_j}\left(\mu\left(\frac{\partial u_i}{\partial x_j} + \frac{\partial u_j}{\partial x_i}\right) + \lambda\delta_{ij}\frac{\partial u_k}{\partial x_k} - \delta_{ij}(3\lambda + 2\mu)\alpha\bar{T}\right) = 0$$

After arranging the set of equations is obtained as follows

$$
\begin{aligned}
0 = {} & \frac{\partial \mu}{\partial x_j}\left(\frac{\partial u_i}{\partial x_j} + \frac{\partial u_j}{\partial x_i}\right) + \mu \frac{\partial^2 u_i}{\partial x_j^2} + \mu \frac{\partial^2 u_j}{\partial x_j x_i} + \frac{\partial \lambda}{\partial x_i}\frac{\partial u_k}{\partial x_k} + \lambda \frac{\partial}{\partial x_i}\left(\frac{\partial u_k}{\partial x_k}\right) \\
& - \left(3\frac{\partial \lambda}{\partial x_i} + 2\frac{\partial \mu}{\partial x_i}\right)\alpha\bar{T} - (3\lambda + 2\mu)\frac{\partial \alpha}{\partial x_i}\bar{T}
\end{aligned}
\tag{16}
$$

For a two-dimensional problem Equation (16) presents a set of two partial differential equations ($i = 1, 2$) as

$$
\begin{aligned}
0 = {} & 2\frac{\partial \mu}{\partial x_1}\frac{\partial u_1}{\partial x_1} + \frac{\partial \mu}{\partial x_2}\left(\frac{\partial u_1}{\partial x_2} + \frac{\partial u_2}{\partial x_1}\right) + (\lambda + 2\mu)\frac{\partial^2 u_1}{\partial x_1^2} \\
& + (\lambda + \mu)\frac{\partial^2 u_2}{\partial x_1 \partial x_2} + \mu\frac{\partial^2 u_1}{\partial x_2^2} + \frac{\partial \lambda}{\partial x_1}\left(\frac{\partial u_1}{\partial x_1} + \frac{\partial u_2}{\partial x_2}\right) \\
& - \left(3\frac{\partial \lambda}{\partial x_1} + 2\frac{\partial \mu}{\partial x_1}\right)\alpha\bar{T} - (3\lambda + 2\mu)\frac{\partial \alpha}{\partial x_1}\bar{T},
\end{aligned}
\tag{17}
$$

$$
\begin{aligned}
0 = {} & \frac{\partial \mu}{\partial x_1}\left(\frac{\partial u_2}{\partial x_1} + \frac{\partial u_1}{\partial x_2}\right) + 2\frac{\partial \mu}{\partial x_2}\frac{\partial u_2}{\partial x_2} + (\lambda + 2\mu)\frac{\partial^2 u_2}{\partial x_2^2} \\
& + (\lambda + \mu)\frac{\partial^2 u_1}{\partial x_1 \partial x_2} + \mu\frac{\partial^2 u_2}{\partial x_1^2} + \frac{\partial \lambda}{\partial x_2}\left(\frac{\partial u_1}{\partial x_1} + \frac{\partial u_2}{\partial x_2}\right) \\
& - \left(3\frac{\partial \lambda}{\partial x_2} + 2\frac{\partial \mu}{\partial x_2}\right)\alpha\bar{T} - (3\lambda + 2\mu)\frac{\partial \alpha}{\partial x_2}\bar{T}
\end{aligned}
\tag{18}
$$

Let $x = x_1$, $y = x_2$, $u(x,y) = u_1(x_1, x_2)$ and $v(x,y) = u_2(x_1, x_2)$ for convenience. Equations (17) and (18) become

$$
\begin{aligned}
0 = {} & 2\frac{\partial \mu}{\partial x}\frac{\partial u}{\partial x} + \frac{\partial \mu}{\partial y}\left(\frac{\partial u}{\partial y} + \frac{\partial v}{\partial x}\right) + (\lambda + 2\mu)\frac{\partial^2 u}{\partial y} \\
& + (\lambda + \mu)\frac{\partial^2 v}{\partial x \partial y} + \mu\frac{\partial^2 u}{\partial y^2} + \frac{\partial \lambda}{\partial x}\left(\frac{\partial u}{\partial x} + \frac{\partial v}{\partial y}\right) \\
& - \left(3\frac{\partial \lambda}{\partial x} + 2\frac{\partial \mu}{\partial x}\right)\alpha\bar{T} - (3\lambda + 2\mu)\frac{\partial \alpha}{\partial x}\bar{T},
\end{aligned}
\tag{19}
$$

$$
\begin{aligned}
0 = {} & \frac{\partial \mu}{\partial x}\left(\frac{\partial v}{\partial x} + \frac{\partial u}{\partial y}\right) + 2\frac{\partial \mu}{\partial y}\frac{\partial v}{\partial y} + (\lambda + 2\mu)\frac{\partial^2 v}{\partial y^2} \\
& + (\lambda + \mu)\frac{\partial^2 u}{\partial x \partial y} + \mu\frac{\partial^2 v}{\partial x^2} + \frac{\partial \lambda}{\partial y}\left(\frac{\partial u}{\partial x} + \frac{\partial v}{\partial y}\right) \\
& - \left(3\frac{\partial \lambda}{\partial y} + 2\frac{\partial \mu}{\partial y}\right)\alpha\bar{T} - (3\lambda + 2\mu)\frac{\partial \alpha}{\partial y}\bar{T}
\end{aligned}
\tag{20}
$$

### 3.1. Boundary Conditions

Boundary conditions are described (Figure 1) as

$$
\bar{T}(x,y) = T_0 \quad \text{constant,}
\tag{21}
$$

$$
v = 0 \quad \text{at } A(0,0) \text{ and } B(0,L),
\tag{22}
$$

$$
\sigma_{xx} = 0, \quad \sigma_{xy} = 0 \quad (x = 0, \ 0 < y < H),
\tag{23}
$$

$$
\sigma_{xx} = 0, \quad \sigma_{xy} = 0 \quad (x = L, \ 0 < y < H),
\tag{24}
$$

$$
\sigma_{yy} = 0, \quad \sigma_{yx} = 0 \quad (y = 0, \ 0 < x < L),
\tag{25}
$$

$$
\sigma_{yy} = 0, \quad \sigma_{yx} = 0 \quad (y = H, \ 0 < x < L),
\tag{26}
$$

$$\sigma_{xx} = 0, \quad \sigma_{yy} = 0, \quad \sigma_{xy} = 0 \quad [A(0,0), B(L,0), G(0,H), F(L,H)], \tag{27}$$

respectively.

### 3.2. Finite Difference Discretization

Let $\psi = \psi(x, y)$ be a continuous, differentiable two-variable function and be defined at each grid point $(i, j)$ with spatial coordinates $(x_i, y_i)$ of a two-dimensional region $\Re$ with a uniform grid distribution (Figure 3). $i$ and $j$ indicate positions along the coordinate axes $x$ and $y$, respectively. The first and second-order partial derivatives of the function $\psi = \psi(x, y)$ with respect to spatial variables $x$ and $y$ can be discretized by means of forward, backward, and central difference equations.

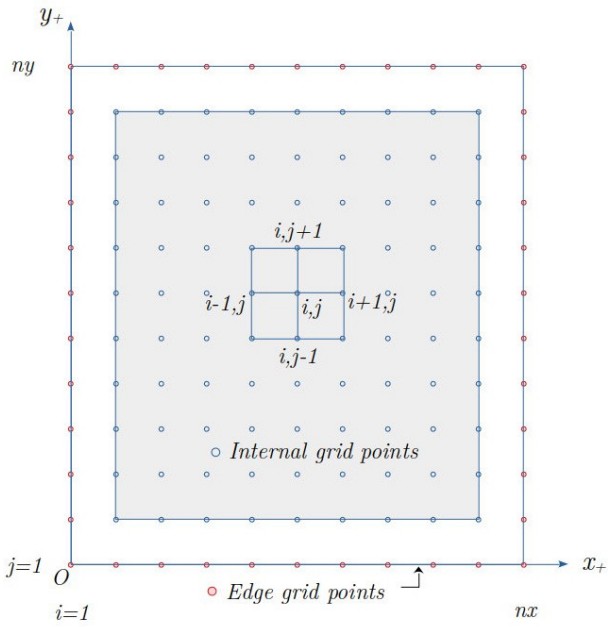

**Figure 3.** Representation of grid-point distributions in the solution region $\Re$ of adhesive single lap joint (not scaled).

The central-difference operators of the first-order partial derivatives of a function $\psi = \psi(x, y)$ at an internal grid point $(i, j)$ with respect to spatial variables $x$ and $y$ are

$$\nabla_x \psi_{i,j} = \left.\frac{\partial \psi}{\partial x}\right|_{i,j} = \frac{\psi_{i+1,j} - \psi_{i-1,j}}{2\Delta x}, \tag{28}$$

$$\nabla_y \psi_{i,j} = \left.\frac{\partial \psi}{\partial y}\right|_{i,j} = \frac{\psi_{i,j+1} - \psi_{i,j-1}}{2\Delta y} \tag{29}$$

As the boundary conditions are applied, the central-difference operators can be modified in a forward sense as

$$\vec{\Gamma}_x \psi_{i,j} = \left.\frac{\partial \psi}{\partial x}\right|_{i,j} = \frac{-3\psi_{i,j} + 4\psi_{i+1,j} - \psi_{i+2,j}}{2\Delta x}, \tag{30}$$

$$\vec{\Gamma}_y \psi_{i,j} = \left.\frac{\partial \psi}{\partial y}\right|_{i,j} = \frac{-3\psi_{i,j} + 4\psi_{i,j+1} - \psi_{i,j+2}}{2\Delta y}, \tag{31}$$

or in a backward sense as

$$\overleftarrow{\Gamma}_x \psi_{i,j} = \left.\frac{\partial \psi}{\partial x}\right|_{i,j} = \frac{3\psi_{i,j} - 4\psi_{i-1,j} + \psi_{i-2,j}}{2\Delta x}, \tag{32}$$

$$\overleftarrow{\Gamma}_y \, \psi_{i,j} = \left. \frac{\partial \psi}{\partial y} \right|_{i,j} = \frac{3\psi_{i,j} - 4\psi_{i,j-1} + \psi_{i,j-2}}{2\Delta y} \tag{33}$$

The central-difference operators of the second-order partial derivatives of a function $\psi = \psi(x, y)$ at a internal grid point $(i, j)$ with respect to spatial variables $x$ and $y$ are

$$\nabla_{xx} \, \psi_{i,j} = \left. \frac{\partial^2 \psi}{\partial x^2} \right|_{i,j} = \frac{\psi_{i+1,j} - 2\psi_{i,j} + \psi_{i-1,j}}{(\Delta x)^2}, \tag{34}$$

$$\nabla_{yy} \, \psi_{i,j} = \left. \frac{\partial^2 \psi}{\partial y^2} \right|_{i,j} = \frac{\psi_{i,j+1} - 2\psi_{i,j} + \psi_{i,j-1}}{(\Delta y)^2}, \tag{35}$$

$$\nabla_{xy} \, \psi_{i,j} = \left. \frac{\partial^2 \psi}{\partial x \partial y} \right|_{i,j} = \frac{\psi_{i+1,j+1} - \psi_{i+1,j-1} - \psi_{i-1,j+1} + \psi_{i-1,j-1}}{4\Delta x \Delta y} \tag{36}$$

As the boundary conditions are applied, the central-difference operators can be modified in a forward or backward sense as

$$\overrightarrow{\Gamma}_{xx} \, \psi_{i,j} = \left. \frac{\partial^2 \psi}{\partial x^2} \right|_{i,j} = \frac{-\psi_{i+3,j} + 4\psi_{i+2,j} - 5\psi_{i+1,j} + 2\psi_{i,j}}{(\Delta x)^2}, \tag{37}$$

$$\overleftarrow{\Gamma}_{xx} \, \psi_{i,j} = \left. \frac{\partial^2 \psi}{\partial x^2} \right|_{i,j} = \frac{2\psi_{i,j} - 5\psi_{i-1,j} + 4\psi_{i-2,j} - \psi_{i-3,j}}{(\Delta x)^2}, \tag{38}$$

$$\overrightarrow{\Gamma}_{yy} \, \psi_{i,j} = \left. \frac{\partial^2 \psi}{\partial y^2} \right|_{i,j} = \frac{-\psi_{i,j+3} + 4\psi_{i,j+2} - 5\psi_{i,j+1} + 2\psi_{i,j}}{(\Delta y)^2}, \tag{39}$$

$$\overleftarrow{\Gamma}_{yy} \, \psi_{i,j} = \left. \frac{\partial^2 \psi}{\partial y^2} \right|_{i,j} = \frac{2\psi_{i,j} - 5\psi_{i,j-1} + 4\psi_{i,j-2} - \psi_{i,j-3}}{(\Delta y)^2} \tag{40}$$

### 3.3. Internal Grid Points

Navier partial differential Equations (19) and (20) at each internal grid point, $(i, j)$ of coordinates $(x_i, y_i)$ inside the region $\Re$ (Figure 3) can be reduced to the linear difference equations by applying the relevant difference operators to the first and second order derivatives as follows

$$
\begin{aligned}
0 = {} & 2\big(\nabla_x \mu_{i,j}\big) \big(\nabla_x u_{i,j}\big) + \big(\nabla_y \mu_{i,j}\big)\big(\nabla_y u_{i,j} + \nabla_x v_{i,j}\big) \\
& + (\lambda + 2\mu)_{i,j}\big(\nabla_{xx} u_{i,j}\big) + (\lambda + \mu)_{i,j}\big(\nabla_{xy} v_{i,j}\big) \\
& + \mu_{i,j}\big(\nabla_{yy} u_{i,j}\big) + \big(\nabla_x \lambda_{i,j}\big)\big(\nabla_x u_{i,j} + \nabla_y v_{i,j}\big) \\
& - \big(3 \nabla_x \lambda_{i,j} + 2 \nabla_x \mu_{i,j}\big)\alpha_{i,j}\bar{T}_{i,j} - (3\lambda + 2\mu)_{i,j}\big(\nabla_x \alpha_{i,j}\big)\bar{T}_{i,j},
\end{aligned}
\tag{41}
$$

$$
\begin{aligned}
0 = {} & \big(\nabla_x \mu_{i,j}\big)\big(\nabla_x v_{i,j} + \nabla_y u_{i,j}\big) + 2\big(\nabla_y \mu_{i,j}\big)\big(\nabla_y v_{i,j}\big) \\
& + (\lambda + 2\mu)_{i,j}\big(\nabla_{yy} v_{i,j}\big) + \mu_{i,j}\big(\nabla_{xx} v_{i,j}\big) \\
& + (\lambda + \mu)_{i,j}\big(\nabla_{xy} u_{i,j}\big) + \big(\nabla_y \lambda_{i,j}\big)\big(\nabla_x u_{i,j} + \nabla_y v_{i,j}\big) \\
& - \big(3 \nabla_y \lambda_{i,j} + 2 \nabla_y \mu_{i,j}\big)\alpha_{i,j}\bar{T}_{i,j} - (3\lambda + 2\mu)_{i,j}\big(\nabla_y \alpha_{i,j}\big)\bar{T}_{i,j}
\end{aligned}
\tag{42}
$$

### 3.4. Boundary Grid Points

Boundary conditions (22)–(27) at each grid point $(i, j)$ of coordinates $(x_i, y_i)$ along the outer boundaries of the region $\Re$ (Figures 1 and 3) can be implemented by means of the difference equations of the stress components

$$\sigma_{xx} = (\lambda + 2\mu)\frac{\partial u}{\partial x} + \lambda\frac{\partial v}{\partial y} - (3\lambda + 2\mu)\alpha\bar{T}, \tag{43}$$

$$\sigma_{yy} = (\lambda + 2\mu)\frac{\partial v}{\partial y} + \lambda \frac{\partial u}{\partial x} - (3\lambda + 2\mu)\alpha \bar{T}, \tag{44}$$

$$\sigma_{xy} = \sigma_{yx} = \mu \left( \frac{\partial u}{\partial y} + \frac{\partial v}{\partial x} \right), \tag{45}$$

as follows

1. Along the edge AG (Equation (23)):

$$0 = (\lambda + 2\mu)_{i,j} \left( \overrightarrow{\Gamma}_x u_{i,j} \right) + \lambda_{i,j} (\nabla_y v_{i,j}) - (3\lambda + 2\mu)_{i,j} \alpha_{i,j} \bar{T}_{i,j}, \tag{46}$$

$$0 = \mu_{i,j} \left( \nabla_y u_{i,j} + \overrightarrow{\Gamma}_x v_{i,j} \right), \tag{47}$$

2. Along the edge BF (Equation (24)):

$$0 = (\lambda + 2\mu)_{i,j} \left( \overleftarrow{\Gamma}_x u_{i,j} \right) + \lambda_{i,j} (\nabla_y v_{i,j}) - (3\lambda + 2\mu)_{i,j} \alpha_{i,j} \bar{T}_{i,j}, \tag{48}$$

$$0 = \mu_{i,j} \left( \nabla_y u_{i,j} + \overleftarrow{\Gamma}_x v_{i,j} \right), \tag{49}$$

3. Along the edge AB (Equation (25)):

$$0 = (\lambda + 2\mu)_{i,j} \left( \overrightarrow{\Gamma}_y v_{i,j} \right) + \lambda_{i,j} (\nabla_x u_{i,j}) - (3\lambda + 2\mu)_{i,j} \alpha_{i,j} \bar{T}_{i,j}, \tag{50}$$

$$0 = \mu_{i,j} \left( \overrightarrow{\Gamma}_y u_{i,j} + \nabla_x v_{i,j} \right), \tag{51}$$

4. Along the edge GF (Equation (26)):

$$0 = (\lambda + 2\mu)_{i,j} \left( \overleftarrow{\Gamma}_y v_{i,j} \right) + \lambda_{i,j} (\nabla_x u_{i,j}) - (3\lambda + 2\mu)_{i,j} \alpha_{i,j} \bar{T}_{i,j}, \tag{52}$$

$$0 = \mu_{i,j} \left( \overleftarrow{\Gamma}_y u_{i,j} + \nabla_x v_{i,j} \right), \tag{53}$$

5. At the corner A (Equation (27)):

$$0 = (\lambda + 2\mu)_{i,j} \left( \overrightarrow{\Gamma}_x u_{i,j} \right) + \lambda_{i,j} \left( \overrightarrow{\Gamma}_y v_{i,j} \right) - (3\lambda + 2\mu)_{i,j} \alpha_{i,j} \bar{T}_{i,j}, \tag{54}$$

$$0 = (\lambda + 2\mu)_{i,j} \left( \overrightarrow{\Gamma}_y v_{i,j} \right) + \lambda_{i,j} \left( \overrightarrow{\Gamma}_x u_{i,j} \right) - (3\lambda + 2\mu)_{i,j} \alpha_{i,j} \bar{T}_{i,j}, \tag{55}$$

$$0 = \mu_{i,j} \left( \overrightarrow{\Gamma}_y u_{i,j} + \overrightarrow{\Gamma}_x v_{i,j} \right), \tag{56}$$

6. At the corner G (Equation (27)):

$$0 = (\lambda + 2\mu)_{i,j} \left( \overrightarrow{\Gamma}_x u_{i,j} \right) + \lambda_{i,j} \left( \overleftarrow{\Gamma}_y v_{i,j} \right) - (3\lambda + 2\mu)_{i,j} \alpha_{i,j} \bar{T}_{i,j}, \tag{57}$$

$$0 = (\lambda + 2\mu)_{i,j} \left( \overleftarrow{\Gamma}_y v_{i,j} \right) + \lambda_{i,j} \left( \overrightarrow{\Gamma}_x u_{i,j} \right) - (3\lambda + 2\mu)_{i,j} \alpha_{i,j} \bar{T}_{i,j}, \tag{58}$$

$$0 = \mu_{i,j} \left( \overleftarrow{\Gamma}_y u_{i,j} + \overrightarrow{\Gamma}_x v_{i,j} \right), \tag{59}$$

7. At the corner B (Equation (27)):

$$0 = (\lambda + 2\mu)_{i,j} \left( \overleftarrow{\Gamma}_x u_{i,j} \right) + \lambda_{i,j} \left( \overrightarrow{\Gamma}_y v_{i,j} \right) - (3\lambda + 2\mu)_{i,j} \alpha_{i,j} \bar{T}_{i,j}, \tag{60}$$

$$0 = (\lambda + 2\mu)_{i,j} \left( \overrightarrow{\Gamma}_y v_{i,j} \right) + \lambda_{i,j} \left( \overleftarrow{\Gamma}_x u_{i,j} \right) - (3\lambda + 2\mu)_{i,j} \alpha_{i,j} \bar{T}_{i,j}, \tag{61}$$

$$0 = \mu_{i,j} \left( \overrightarrow{\Gamma}_y u_{i,j} + \overleftarrow{\Gamma}_x v_{i,j} \right), \tag{62}$$

8. At the corner F (Equation (27)):

$$0 = (\lambda + 2\mu)_{i,j} \left( \overleftarrow{\Gamma}_x u_{i,j} \right) + \lambda_{i,j} \left( \overleftarrow{\Gamma}_y v_{i,j} \right) - (3\lambda + 2\mu)_{i,j} \alpha_{i,j} \bar{T}_{i,j}, \tag{63}$$

$$0 = (\lambda + 2\mu)_{i,j} \left( \overleftarrow{\Gamma}_y v_{i,j} \right) + \lambda_{i,j} \left( \overleftarrow{\Gamma}_x u_{i,j} \right) - (3\lambda + 2\mu)_{i,j} \alpha_{i,j} \bar{T}_{i,j}, \tag{64}$$

$$0 = \mu_{i,j} \left( \overleftarrow{\Gamma}_y u_{i,j} + \overleftarrow{\Gamma}_x v_{i,j} \right) \tag{65}$$

*3.5. Discretization of Continuity Conditions along Interfaces*

The adherend and adhesive sides of interfaces are of different material properties. In single material regions, the finite difference discretization of Navier equations and boundary conditions can be made easily. The bi-material interfaces result in sudden changes in the through-thickness mechanical and thermal properties of local material. Consequently, the thermal strains and stresses become discontinuous while the total normal strains, as well as displacement components, are continuous.

In the infinitesimal neighborhood of bi-material interfaces, twin grid points can be defined (Figure 3), then Navier equations and boundary conditions can be discretized with material properties of the grid points on both sides $(-, +)$ of interfaces in sequence, and the predicted displacement components can be equated. Thus,

$$u_{i,j}^- = u_{i,j}^+, \tag{66}$$

$$v_{i,j}^- = v_{i,j}^+ \tag{67}$$

3.5.1. Internal Grid Points of Interfaces

The Navier Equation (41) can be discretized with material properties of the internal grid points on both sides $(-, +)$ of interfaces as

$$
\begin{aligned}
0 = {} & 2\left(\nabla_x \mu_{i,j}\right)^- \left(\nabla_x u_{i,j}\right) + \left(\nabla_y \mu_{i,j}\right)^- \left(\nabla_y u_{i,j} + \nabla_x v_{i,j}\right) \\
& + (\lambda + 2\mu)_{i,j}^- \left(\nabla_{xx} u_{i,j}\right) + (\lambda + \mu)_{i,j}^- \left(\nabla_{xy} v_{i,j}\right) \\
& + \mu_{i,j}^- \left(\nabla_{yy} u_{i,j}\right) + \left(\nabla_x \lambda_{i,j}\right)^- \left(\nabla_x u_{i,j} + \nabla_y v_{i,j}\right) \\
& - \left(3 \nabla_x \lambda_{i,j} + 2 \nabla_x \mu_{i,j}\right)^- \alpha_{i,j}^- \bar{T}_{i,j} - (3\lambda + 2\mu)_{i,j}^- \left(\nabla_x \alpha_{i,j}\right)^- \bar{T}_{i,j},
\end{aligned}
\tag{68}
$$

$$
\begin{aligned}
0 = {} & 2\left(\nabla_x \mu_{i,j}\right)^+ \left(\nabla_x u_{i,j}\right) + \left(\nabla_y \mu_{i,j}\right)^+ \left(\nabla_y u_{i,j} + \nabla_x v_{i,j}\right) \\
& + (\lambda + 2\mu)_{i,j}^+ \left(\nabla_{xx} u_{i,j}\right) + (\lambda + \mu)_{i,j}^+ \left(\nabla_{xy} v_{i,j}\right) \\
& + \mu_{i,j}^+ \left(\nabla_{yy} u_{i,j}\right) + \left(\nabla_x \lambda_{i,j}\right)^+ \left(\nabla_x u_{i,j} + \nabla_y v_{i,j}\right) \\
& - \left(3 \nabla_x \lambda_{i,j} + 2 \nabla_x \mu_{i,j}\right)^+ \alpha_{i,j}^+ \bar{T}_{i,j} - (3\lambda + 2\mu)_{i,j}^+ \left(\nabla_x \alpha_{i,j}\right)^+ \bar{T}_{i,j},
\end{aligned}
\tag{69}
$$

respectively. In Equations (68) and (69), $j + 1$ is replaced by $j + 2$, and $j - 1$ by $j - 2$ in all finite difference operators for the grid points along the lower and upper twins of interfaces, respectively.

Similarly, the Navier Equation (42) can be discretized with material properties of the internal grid points on the both sides $(-, +)$ of interfaces as

$$
\begin{aligned}
0 = {} & \left(\nabla_x \mu_{i,j}\right)^- \left(\nabla_x v_{i,j} + \nabla_y u_{i,j}\right) + 2\left(\nabla_y \mu_{i,j}\right)^- \left(\nabla_y v_{i,j}\right) \\
& + (\lambda + 2\mu)_{i,j}^- \left(\nabla_{yy} v_{i,j}\right) + \mu_{i,j}^- \left(\nabla_{xx} v_{i,j}\right) \\
& + (\lambda + \mu)_{i,j}^- \left(\nabla_{xy} u_{i,j}\right) + \left(\nabla_y \lambda_{i,j}\right)^- \left(\nabla_x u_{i,j} + \nabla_y v_{i,j}\right) \\
& - \left(3 \nabla_y \lambda_{i,j} + 2 \nabla_y \mu_{i,j}\right)^- \alpha_{i,j}^- \bar{T}_{i,j} - (3\lambda + 2\mu)_{i,j}^- \left(\nabla_y \alpha_{i,j}\right)^- \bar{T}_{i,j},
\end{aligned}
\tag{70}
$$

$$
\begin{aligned}
0 = {} & \left(\nabla_x \mu_{i,j}\right)^+ \left(\nabla_x v_{i,j} + \nabla_y u_{i,j}\right) + 2\left(\nabla_y \mu_{i,j}\right)^+ \left(\nabla_y v_{i,j}\right) \\
& + (\lambda + 2\mu)^+_{i,j}\left(\nabla_{yy} v_{i,j}\right) + \mu^+_{i,j}\left(\nabla_{xx} v_{i,j}\right) \\
& + (\lambda + \mu)^+_{i,j}\left(\nabla_{xy} u_{i,j}\right) + \left(\nabla_y \lambda_{i,j}\right)^+ \left(\nabla_x u_{i,j} + \nabla_y v_{i,j}\right) \\
& - \left(3\nabla_y \lambda_{i,j} + 2\nabla_y \mu_{i,j}\right)^+ \alpha^+_{i,j}\bar{T}_{i,j} - (3\lambda + 2\mu)^+_{i,j}\left(\nabla_y \alpha_{i,j}\right)^+ \bar{T}_{i,j}
\end{aligned}
\tag{71}
$$

The displacement components $u_{i,j}$ and $v_{i,j}$ can be calculated at each twin node in the infinitesimal neighborhood of both interfaces.

### 3.5.2. Grid Points at the Free Edges of Interfaces

The boundary conditions at the grid points along the free edges AG ($x = 0$) and BF ($x = L$) of both interfaces can be implemented with the previous approach as

1. At the free edge AG of both interfaces:

$$
\sigma^-_{xx} = 0 = (\lambda + 2\mu)^-_{i,j}\left(\overrightarrow{\Gamma}_x u_{i,j}\right) + \lambda^-_{i,j}\left(\nabla_y v_{i,j}\right) - (3\lambda + 2\mu)^-_{i,j}\alpha^-_{i,j}\bar{T}_{i,j},
\tag{72}
$$

$$
\sigma^+_{xx} = 0 = (\lambda + 2\mu)^+_{i,j}\left(\overrightarrow{\Gamma}_x u_{i,j}\right) + \lambda^+_{i,j}\left(\nabla_y v_{i,j}\right) - (3\lambda + 2\mu)^+_{i,j}\alpha^+_{i,j}\bar{T}_{i,j},
\tag{73}
$$

$$
\sigma^-_{xy} = 0 = \mu^-_{i,j}\left(\nabla_y u_{i,j} + \overrightarrow{\Gamma}_x v_{i,j}\right),
\tag{74}
$$

$$
\sigma^+_{xy} = 0 = \mu^+_{i,j}\left(\nabla_y u_{i,j} + \overrightarrow{\Gamma}_x v_{i,j}\right),
\tag{75}
$$

2. At the edge BF of both interfaces:

$$
\sigma^-_{xx} = 0 = (\lambda + 2\mu)^-_{i,j}\left(\overleftarrow{\Gamma}_x u_{i,j}\right) + \lambda^-_{i,j}\left(\nabla_y v_{i,j}\right) - (3\lambda + 2\mu)^-_{i,j}\alpha^-_{i,j}\bar{T}_{i,j},
\tag{76}
$$

$$
\sigma^+_{xx} = 0 = (\lambda + 2\mu)^+_{i,j}\left(\overleftarrow{\Gamma}_x u_{i,j}\right) + \lambda^+_{i,j}\left(\nabla_y v_{i,j}\right) - (3\lambda + 2\mu)^+_{i,j}\alpha^+_{i,j}\bar{T}_{i,j},
\tag{77}
$$

$$
\sigma^-_{xy} = 0 = \mu^-_{i,j}\left(\nabla_y u_{i,j} + \overleftarrow{\Gamma}_x v_{i,j}\right),
\tag{78}
$$

$$
\sigma^+_{xy} = 0 = \mu^+_{i,j}\left(\nabla_y u_{i,j} + \overleftarrow{\Gamma}_x v_{i,j}\right)
\tag{79}
$$

### 3.6. Solution Method

After Navier equations and the boundary conditions are discretized in the solution region $\Re$, unknown displacement components $u_{i,j}$ and $v_{i,j}$ at each grid point $(i, j)$ can be calculated by implementing a recursive error reducing method.

Let $u^k_{i,j}$ and $v^k_{i,j}$ be displacement components at an iteration index $k$. The finite difference representations for Navier Equations (41)–(42) and (68)–(71) and boundary conditions (46)–(65) and (72)–(79) can be written as

$$
\begin{aligned}
c_{i,j} u^k_{i,j} &= F^k_{i,j}, \\
d_{i,j} v^k_{i,j} &= G^k_{i,j}
\end{aligned}
\tag{80}
$$

The error levels for iteration $k$ are calculated as

$$
\begin{aligned}
\mathrm{erru}^k_{i,j} &= F^k_{i,j} - \frac{1}{c_{i,j}} u^k_{i,j}, \\
\mathrm{errv}^k_{i,j} &= G^k_{i,j} - \frac{1}{d_{i,j}} v^k_{i,j}
\end{aligned}
\tag{81}
$$

The displacement components for iteration $k + 1$ are predicted as

$$
\begin{aligned}
u^{k+1}_{i,j} &= u^k_{i,j} - \frac{\mathrm{erru}^k_{i,j}}{c_{i,j}}, \\
v^{k+1}_{i,j} &= v^k_{i,j} - \frac{\mathrm{errv}^k_{i,j}}{d_{i,j}}
\end{aligned}
\tag{82}
$$

The summation of errors is calculated as

$$\text{SumError} = \sum_{i=1}^{m} \sum_{j=1}^{n} \left( \text{erru}_{i,j}^{k} + \text{errv}_{i,j}^{k} \right) \tag{83}$$

where $m$ and $n$ are division numbers of the uniform grid for the region $\Re$ along the coordinate axes $x$ and $y$, respectively. The loop between (80) and (83) is repeated until SumError is reduced to a specific error level of eps $= 10^{-8}$ by equating $u_{i,j}^{k+1}$ and $v_{i,j}^{k+1}$ to $u_{i,j}^{k}$ and $v_{i,j}^{k}$.

## 4. Results and Discussion

The geometry, dimensions, and boundary conditions of an aluminum single lap joint bonded with a functionally graded adhesive are shown in Figure 1. The joint length $L = 15$ mm, aluminum adherend thickness $t_1 = t_3 = 1.5$ mm, and adhesive thickness $t_2 = 0.5$ mm. The displacements of the left and right lower corners are fixed only in the $y-$direction, and the normal and shear stresses are considered as zero along the free edges of the adhesive joint. A constant temperature change of $\Delta T = 30\ °C$ is assumed at all grid points. The thermal and mechanical properties of adherend, micro-sized powder, and adhesive materials are given in Table 1. The solution domain was discretized into a mesh grid of $631 \times 150$ with increments of $dx = dy = 0.02381$ mm along the x- and y-directions, respectively.

**Table 1.** Thermal and mechanical properties of the adherend, adhesive, and micro-sized particles.

|  |  | Units | Aluminium | Al$_2$O$_3$ | Epoxy |
|---|---|---|---|---|---|
| Modulus of elasticity | $E$ | MPa | 68,900.0 | 379,211.0 | 4391.43 |
| Poisson's ratio | $v$ |  | 0.33 | 0.19 | 0.34 |
| Coefficient of thermal expansion | $\alpha$ | 1/m·K | $23.58 \times 10^{-6}$ | $7.6 \times 10^{-6}$ | $40.47 \times 10^{-6}$ |
| Shear modulus | $G$ | MPa | 26,000.0 | 150,000.0 | 1638.6 |
| Bulk modulus | $K$ | MPa | 67,549.0 | 203,876.9 | 4574.4 |

The composition of functionally graded adhesive consists of (Al$_2$O$_3$) micro-particles and two-component epoxy-based adhesive mixed at specific volume fractions. The volume fraction of micro-particles $V_p$ through the thickness of the adhesive layer is tailored by obeying a power rule including a gradient power index $n$ (Equation (2)). Two grading directions are also considered, thus, the variations of volume fraction of particles through the adhesive thickness between a particle-rich adhesive around the lower interface and a neat adhesive around the upper interface (PRA→NA, Figure 2a), and between a neat adhesive around the lower interface and a particle-rich adhesive around the upper interface (NA→PRA, Figure 2b), respectively. The volume fraction of particles $V_p$ around the particle-rich adhesive interface is limited by the maximum volume fraction of particles as $V_{max} = 0.01, 0.1, 0.3$ because the adhesion between adhesive and aluminum adherend has not deteriorated. The volume fraction of particles $V_p$ can have a desired through-thickness form with the power index $n = 0.1, 0.2, 0.3, 0.5, 1.0, 2.0, 4.0, 8.0$ and $12.0$ (Figure 2).

Consequently, this study aims to determine the effects of the gradient power index $n$, maximum volume fraction $V_{max}$, and grading direction through the adhesive thickness on the adhesive strain and stress distributions induced by a uniform temperature change through the adhesive joint and on reliving their critical levels. The thermal stress analyses were carried out for various maximum volume fractions of particles and power index values. The strain and stress components exhibited almost similar distributions except for their levels of low-volume fractions of particles. The left and right free edges of the adhesive layer appeared as symmetrical stress and strain concentration regions. Therefore, the distributions of strain and stress components will be discussed via their contour plots.

Figures 4–6 show the effects of the power index ($n$) and the maximum volume fraction of particles ($V_{max}$) on the distributions of strain $\varepsilon_{ij}(x, y)$ and stress $\sigma_{ij}(x, y)$ components around the left free ends of both lower and upper interfaces of a functionally graded

adhesive layer. The normal strain $\varepsilon_{xx}$ concentrates around the lower and upper adhesive interfaces, propagates towards the overlapping center, and decreases uniformly (Figure 4). A graded variation of normal strain appears, especially in the vicinity of both interfaces. The free edges of the adhesive layer undergo higher normal strain levels than two adherends because the overall coefficient of thermal expansion of adhesive composition is larger than those of aluminum adherends. Similar symmetrical distributions also appear around the right-hand sides of both interfaces. The general form of distributions is not affected by the power index $n$ for a low maximum volume fraction ($V_{max} = 0.01$), while its effect becomes more apparent for a through-thickness quite particle-rich composition ($V_{max} = 0.3$). The normal strain levels decrease as the adhesive is enriched by particles through the adhesive thickness ($n = 1.0 \rightarrow 12$). Thus, the higher strain levels occur around the free end of the upper interface and in the vicinity of this interface. The normal strain decreases slightly through the adhesive thickness towards the vicinity of the lower interface. The normal strain distribution exhibits a dependency on the through-thickness variation of particles in the adhesive composition. The normal stress $\sigma_{xx}$ concentrates in the vicinity of both interfaces and exhibits discontinuous distributions at different levels on the adherend and adhesive sides of interfaces (Figure 4). The normal stress levels in the adhesive layer are rather lower than in the adherends because aluminum adherends are of higher modulus than the adhesive composition. The normal stress distribution and levels remain similar for a low maximum volume fraction of particles (0.01), and the effect of the power index is negligible. The power index has a more apparent effect on the normal stress distributions and levels for a high volume fraction of particles (0.3). Namely, the higher stress levels occur on the adherend side in the vicinity of the neat adhesive–upper adherend interface in comparison with those around the particle-rich adhesive–lower adherend interface as the adhesive is enriched by particles through its thickness.

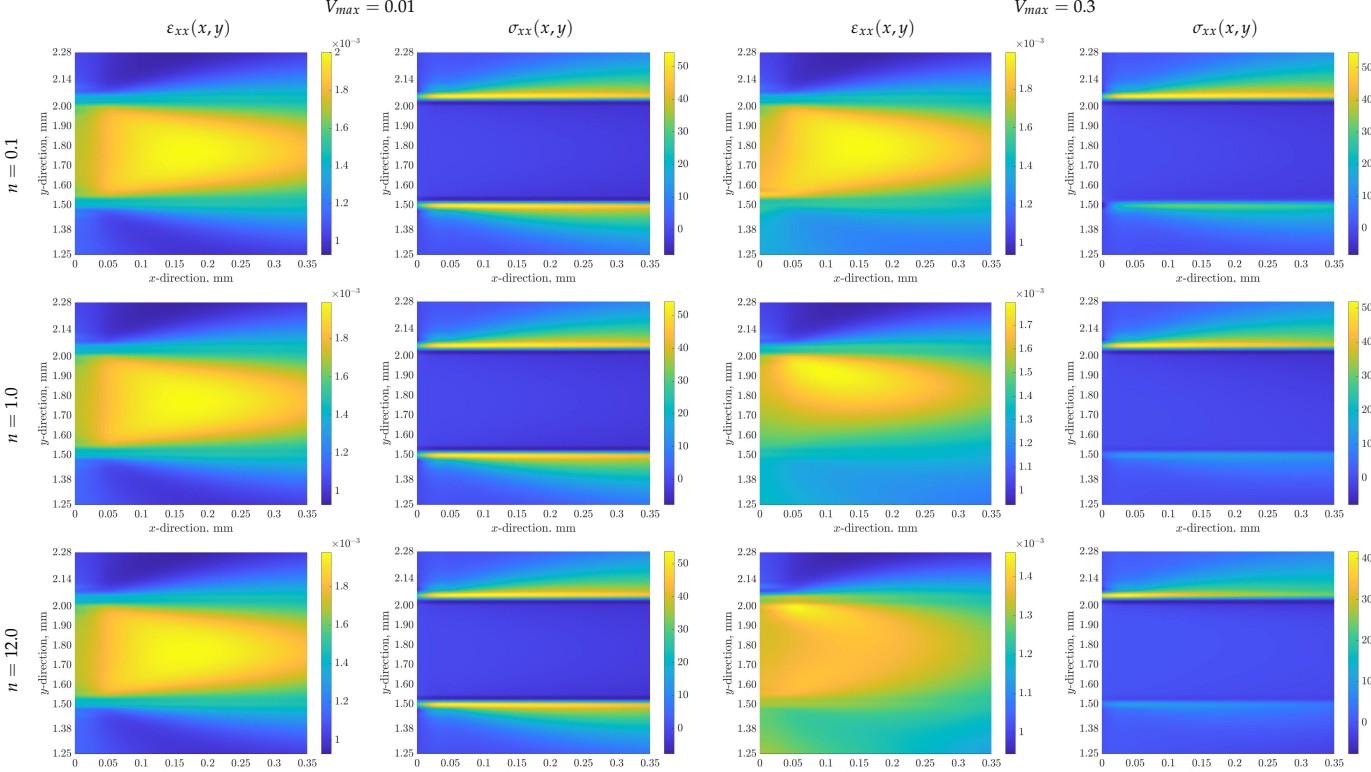

**Figure 4.** Effects of the gradient power index ($n$) and the maximum volume fraction of particles ($V_{max}$) on the normal strain $\varepsilon_{xx}(x, y)$ and stress $\sigma_{xx}(x, y)$ distributions around the left free-end of the PRA → NA functionally graded adhesive layer and interfaces.

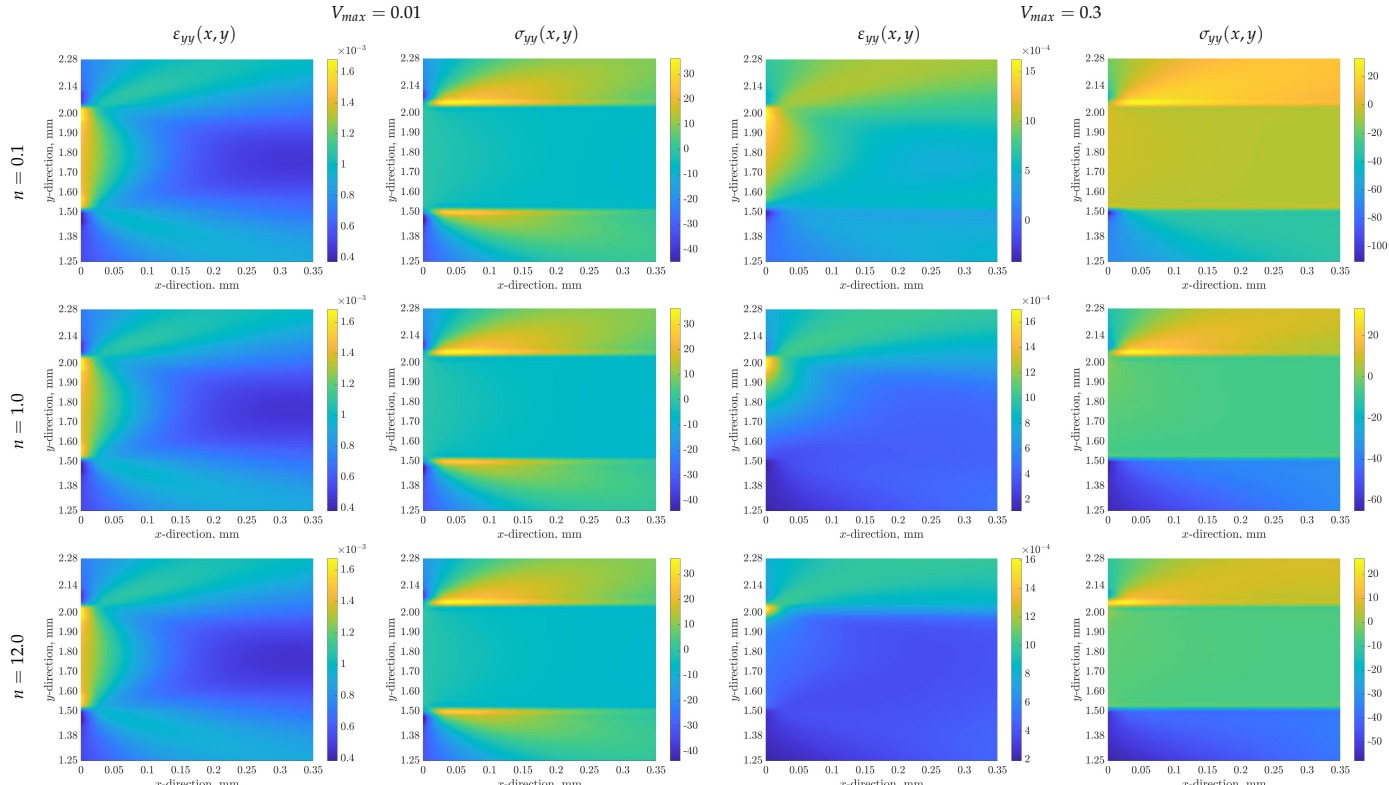

**Figure 5.** Effects of the gradient power index ($n$) and the maximum volume fraction of particles ($V_{max}$) on the normal strain $\varepsilon_{yy}(x, y)$ and stress $\sigma_{yy}(x, y)$ distributions around the left free-end of the PRA → NA functionally graded adhesive layer and interfaces.

The normal strain $\varepsilon_{yy}$ exhibits similar distributions; thus, the concentration zones appear around the free edges of the adhesive layer and both adhesive interfaces, and then continuously relieve through the adherend and adhesive regions neighboring to the interfaces towards the center of overlap region (Figure 5). Especially, the concentration regions around the free ends of the adhesive layer expand through the adhesive thickness towards the center of the overlap region. The normal strain decreases along a symmetrical diffusing band-form region on the adherend sides of two interfaces. The normal strain levels are higher in the adhesive layer because the adhesive layer is of a larger overall coefficient of thermal expansion. The distribution manner and levels of the normal strain $\varepsilon_{yy}$ around the free edges of the adhesive layer, and both interfaces are not affected notably by the power index for a low maximum volume fraction of particles (0.01). The power index becomes more effective on the normal strain distributions and levels for a high volume fraction of particles (0.3). Namely, the strain concentration zone at the adhesive-free edge gets narrower towards the free end of the upper neat adhesive–adherend interface, and occurs in the vicinity of this interface towards the center of the overlap region. In addition, the diffusing zone in the upper adherend becomes more apparent in comparison with those in the lower adherend as the adhesive is enriched by particles through its thickness. The normal stress $\sigma_{yy}$ exhibits concentration zones with discontinuous distributions in the vicinity of both interfaces and different levels on the adherend and adhesive sides of interfaces (Figure 5). The normal stress levels in the adhesive layer are rather lower, whereas the adherend regions neighboring both interfaces undergo higher stress levels diffusing towards the center of the overlap region and through adherend thickness. The normal stress distribution and levels remain similar, and the effect of the power index is negligible for a low maximum volume fraction of particles (0.01). However, the power index affects evidently the normal stress distributions and levels only for a high volume fraction of particles (0.3). The higher tensile stress levels occur on the adherend side in the

vicinity of the neat adhesive–upper adherend interface, whereas the higher compressive stress levels occur on the adherend side in the vicinity of the particle-rich adhesive–lower adherend interface. The high-stress levels relieve considerably in these concentration zones as the adhesive is enriched by particles through its thickness.

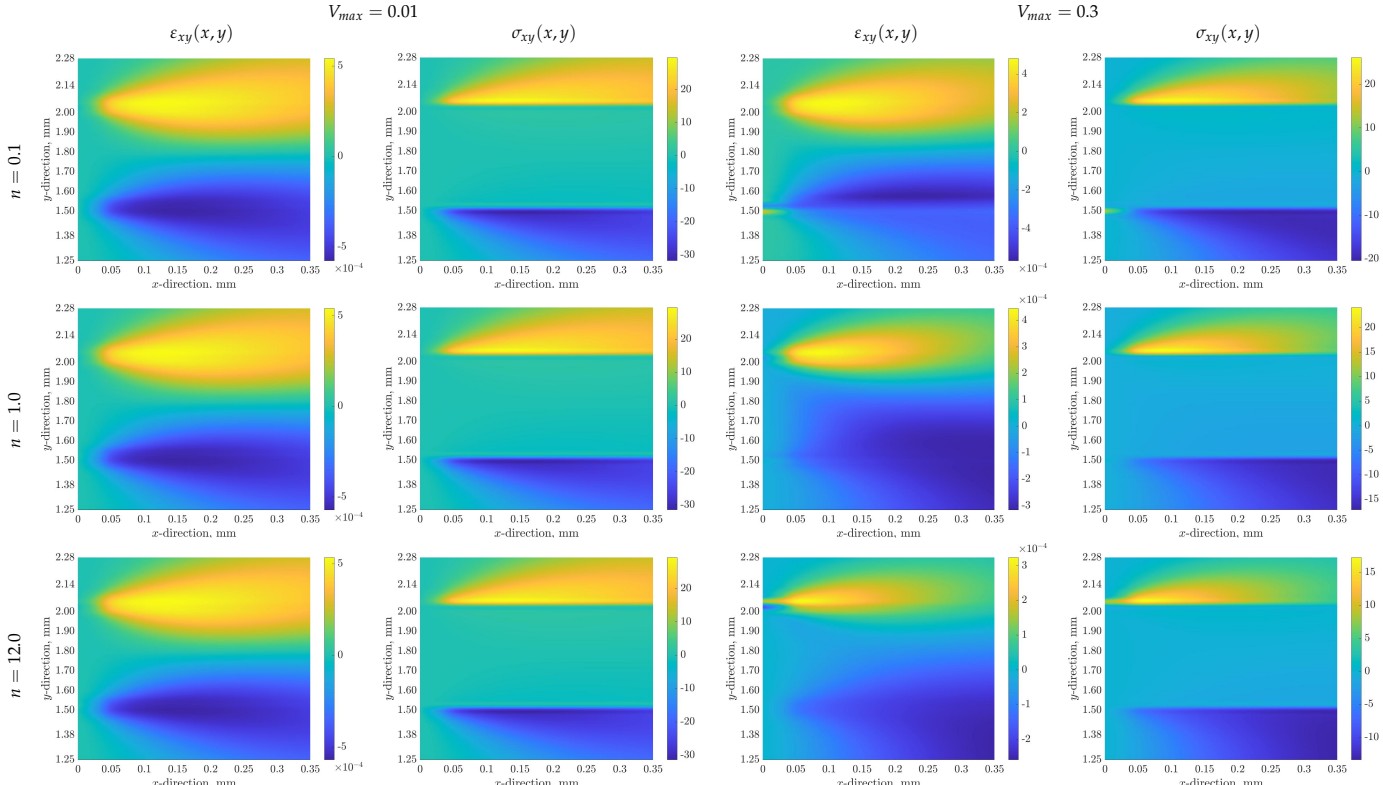

**Figure 6.** Effects of the gradient power index ($n$) and the maximum volume fraction of particles ($V_{max}$) on the shear strain $\varepsilon_{xy}(x,y)$ and stress $\sigma_{xy}(x,y)$ distributions around the left free-end of the PRA $\rightarrow$ NA functionally graded adhesive layer and interfaces.

The shear strain $\varepsilon_{xy}$ concentrates in the neighborhood of both interfaces except their free edges (Figure 6). A narrow concentration zone initiates at a small distance from the free ends of both interfaces and expands through the adherend and adhesive thicknesses. The shear deformations occur symmetrically but in opposite directions along the lower and upper interfaces. The power index has a negligible effect on both distribution manner and levels of the shear strain $\varepsilon_{xy}$ around the free edges of the adhesive layer and both interfaces for a low maximum volume fraction of particles (0.01), whereas this effect becomes apparent on the shear strain distributions and levels for a high volume fraction of particles (0.3). Thus, the symmetrical concentration zones near both interfaces degenerate, and the shear strain levels around the neat adhesive–upper adherend interface reach partly higher levels, and the shear strain levels decrease as the through-thickness adhesive composition is enriched by particles. The neat adhesive interface forces the neighboring zones of both adhesive and adherend to more deformation in shear due to its tendency to a larger thermal expansion. The shear stress $\sigma_{xy}$ concentrations occur in the neighborhood of both interfaces except their free edges and exhibit a discontinuous distribution along both interfaces (Figure 6). The adherend side experiences higher shear stresses than those on the adhesive side. The shear strain and stress distributions are conformable. The effect of the power index becomes apparent on the shear stress $\sigma_{xy}$ distributions and levels in the regions near both interfaces only for a high volume fraction of particles (0.3). The symmetrical concentration zones near both interfaces become smaller towards the free ends of both interfaces, and the shear stress levels decrease as the through-thickness adhesive composition is enriched by particles.

The shear stress is also as critical as the normal stresses. In general, a ductile adhesive composition is expected to tend to damage in shear.

In case an adhesive single lap joint is subjected to a uniform temperature distribution, the strain and stress components concentrate in the neighborhood of adherend–adhesive interfaces and around their free edges. They also decrease uniformly along the interfaces towards the center of the overlap region. The normal and shear strain components exhibit grading distributions along both bi-material interfaces, whereas the normal and shear stresses are of a discontinuous nature. Since the adhesive material is generally assumed to be a material of lower strength than adherends; hereafter, the evaluation of normal/shear strain and stress distributions around the free edges of the adhesive layer is in evidence as a more convenient way.

Figures 7–9 show the effects of both gradient power index ($n$) and maximum volume fraction of particles ($V_{max}$) on the distributions of strain $\varepsilon_{ij}(x, y)$ and stress $\sigma_{ij}(x, y)$ components around the left free end of a functionally graded adhesive layer. The normal strain $\varepsilon_{xx}$ distributions are symmetrical with respect to the adhesive mid-line (Figure 7) for a low maximum volume fraction of particles (0.01). The high normal strain levels occur in a large adhesive zone in the vicinity of adhesive mid-line from at a small distance from the free ends of both interfaces and decrease uniformly towards the center of the overlap region. The power index exhibits a negligible effect on the normal strain distribution and levels for a low maximum volume fraction of particles. Increasing the maximum volume fractions of particles makes the effect of the power index to become more apparent. As the power index is increased, namely, the through-thickness adhesive composition is enriched by particles, the symmetrical distribution of normal strain disappears, and the concentration region contracts towards the neat adhesive–upper adherend interface. This indicates a reduced overall thermal expansion of remaining adhesive regions enriched by particles. The normal stress $\sigma_{xx}$ acts in compression and is negligible near the adhesive-free ends. It increases through a limited expanding adhesive region towards the center of the overlap region in a symmetrical manner with respect to the adhesive mid-line (Figure 7) for a low maximum volume fraction of particles. In addition, the adhesive regions near the interfaces experience higher compressive stress levels. As the local adhesive composition through the adhesive thickness is enriched by particles, the power index has a negligible effect on the normal stress distribution and levels. However, a high maximum volume fraction of particles reveals the effect of the power index. The normal stress distribution is not symmetrical anymore with respect to the adhesive mid-line, and a larger adhesive region undergoes lower stress levels. The adhesive regions near the neat adhesive–adherend interface experience still higher compressive stress levels while the stress levels decrease apparently in the adhesive regions near the lower particle-rich adhesive–adherend interface with increasing power index.

The normal strain $\varepsilon_{yy}$ distributions are also symmetrical with respect to the adhesive mid-line and concentrate around the free ends of the adhesive layer, lower and upper interfaces. It decreases uniformly towards the center of the overlap region (Figure 8). The remaining adhesive regions undergo lower normal strain levels. For a low maximum volume fraction of particles, the power index is of a small effect on the normal strain distribution and levels. The symmetrical distribution with respect to the adhesive mid-line disappears, and the high strain concentration zones distribute from the adhesive mid-line towards the free end of neat adhesive–upper adherend interface for a high maximum volume fraction of particles (0.3). The power index affects also the through-thickness variation of normal strain; thus, the normal strain levels are formed depending on the variation of volume fraction of particles through the adhesive thickness, and the strain concentration zone contracts around the free end of the neat adhesive–upper adherend interface. The adhesive zones near the neat adhesive–upper adherend interface experience higher normal strain levels and decrease uniformly towards the particle-rich adhesive–lower adherend interface. The normal stress $\sigma_{yy}$ distributions (Figure 8) also conform with those of the normal strain $\varepsilon_{yy}$. The adhesive layer undergoes compressive stresses except

for the adhesive-free edge. Symmetrical distribution of normal stress appears with respect to the adhesive mid-line for a low maximum volume fraction of particles. The power index has a negligible effect on both normal stress distribution and levels. However, the lower normal stress zones around the adhesive free end contract around the free end of the neat adhesive–upper adherend interface as the local adhesive composition through the adhesive thickness is enriched by particles for a higher maximum volume fraction of particles (0.3). The power index is more effective on the normal stress distribution and levels. The remaining adhesive regions undergo compressive stresses, which increase uniformly towards the free end of the particle-rich adhesive–lower adherend interface, while the normal stress levels increase slightly because $Al_2O_3$ particles with high modulus improve the overall modulus of the local adhesive composition.

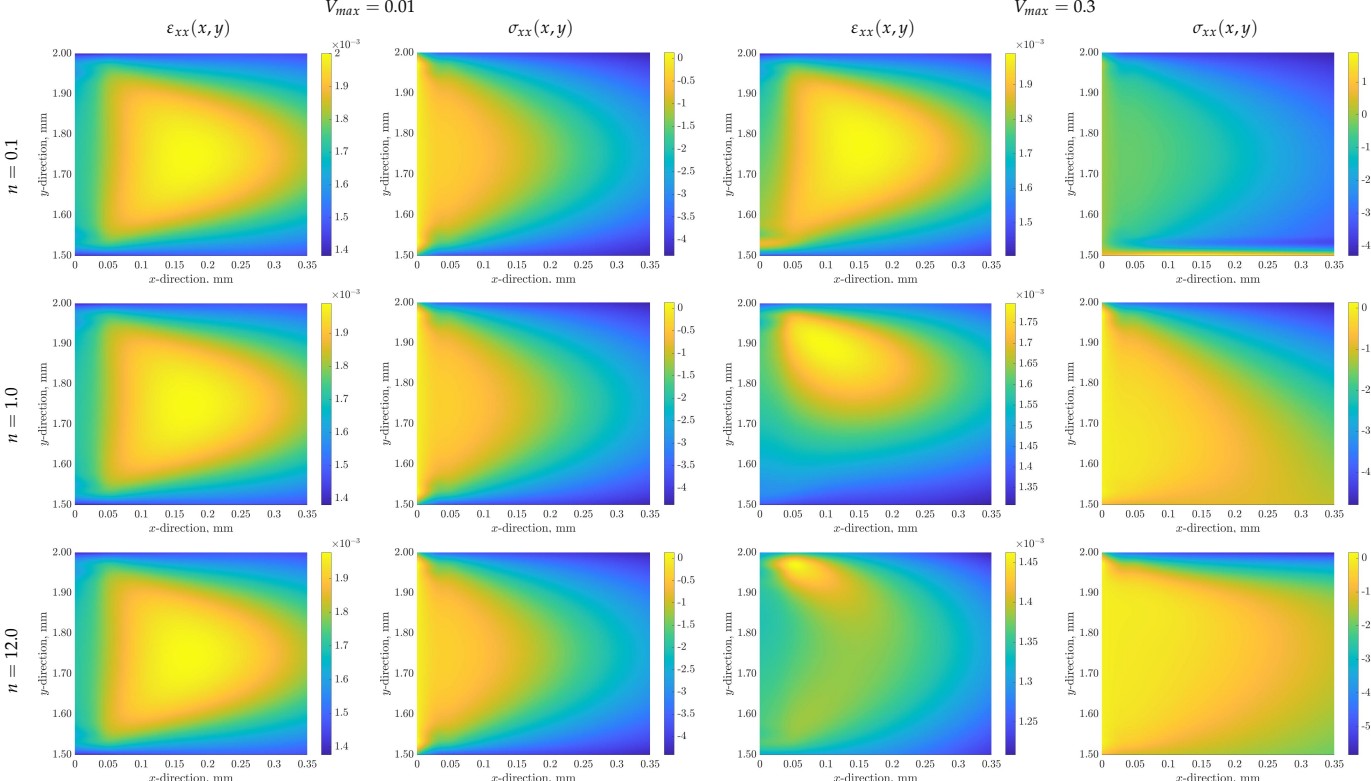

**Figure 7.** Effects of the gradient power index ($n$) and the maximum volume fraction of particles ($V_{max}$) on the normal strain $\varepsilon_{xx}(x,y)$ and stress $\sigma_{xx}(x,y)$ distributions around the left free-end of the PRA $\rightarrow$ NA functionally graded adhesive layer.

The shear strain $\varepsilon_{xy}$ is of a symmetrical distribution with respect to the adhesive mid-line, which concentrates on the free ends of the lower and upper interfaces and expands uniformly through the adhesive regions near both interfaces towards the adhesive mid-line and the center of overlap region (Figure 9). The shear strain acts in opposite directions in the upper and lower adhesive portions. The remaining adhesive regions towards the center of the overlap region undergo negligible shear strain distributions. For a low maximum volume fraction of particles, the power index has a negligible effect on the shear strain distribution and levels, whereas a higher maximum volume fraction of particles (0.3) affects both shear strain distribution and levels. Namely, as the through-thickness adhesive composition is enriched by particles, the symmetrical distribution of shear strain degenerates fully, the high shear strain region in the adhesive upper portion contracts a narrower adhesive region near the upper neat adhesive–adherend interface and expands along the neighborhood of this interface towards the center of the overlap region. The shear strain levels also decrease uniformly, and its distribution is formed according to the variation of particle volume fraction through the adhesive thickness. The shear stress $\sigma_{xy}$

exhibits a symmetrical distribution with respect to the adhesive mid-line, concentrates around the free ends of the lower and upper interfaces, and high shear stress regions expand through the adhesive thickness uniformly towards the adhesive mid-line (Figure 9). The shear stress acts in opposite directions in the upper and lower adhesive portions. A large of the remaining overlap region experiences negligible shear stress levels. The shear stress distributions are less critical than those of two normal stress components. For a low maximum volume fraction of particles (0.01), the power index has a negligible effect on the shear stress distribution and levels. A higher maximum volume fraction of particles results in the effect of the power index becoming more apparent in the shear stress distribution and levels. Namely, the symmetrical shear stress distribution disappears, and the shear stresses in the particle-rich adhesive regions become more apparent. As the adhesive composition is enriched by particles, the shear stress levels decrease uniformly.

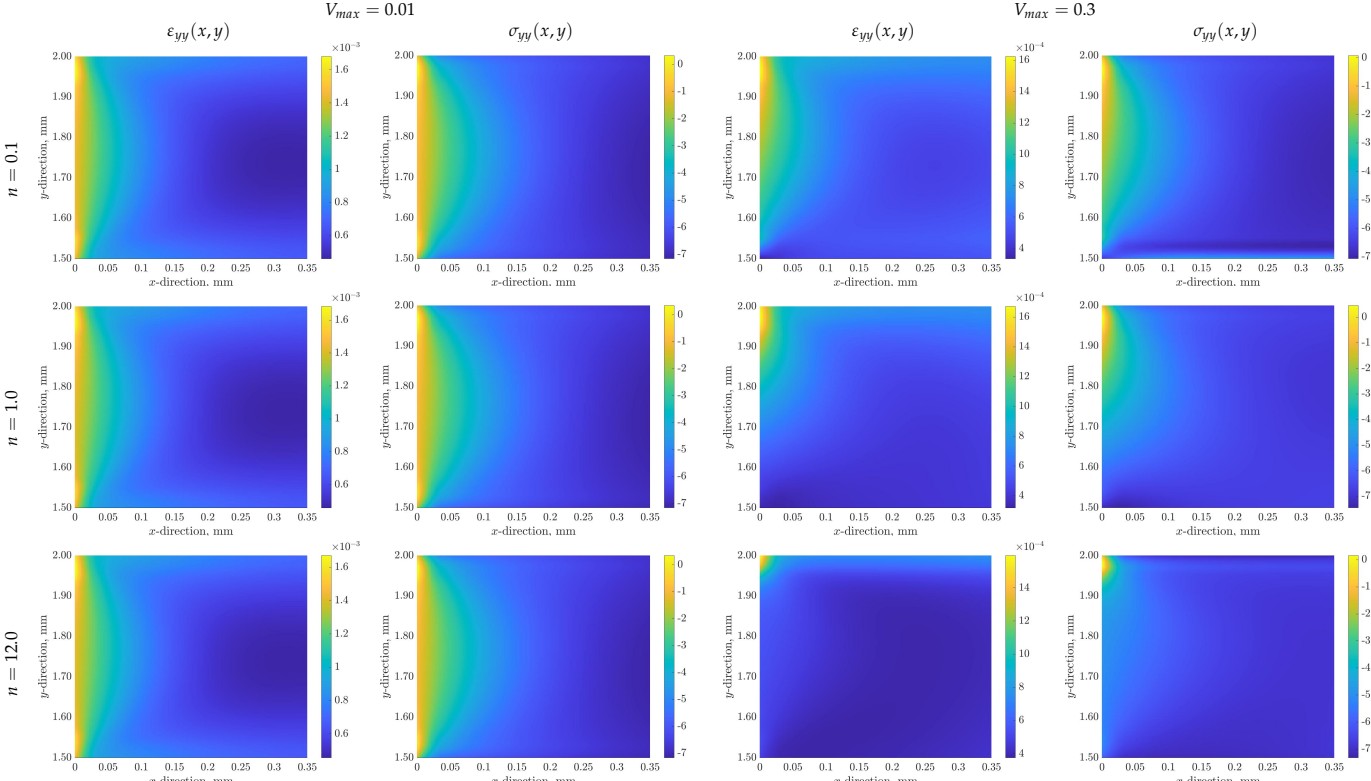

**Figure 8.** Effects of the gradient power index ($n$) and the maximum volume fraction of particles ($V_{max}$) on the normal strain $\varepsilon_{yy}(x, y)$ and stress $\sigma_{yy}(x, y)$ distributions around the left free-end of the PRA $\rightarrow$ NA functionally graded adhesive layer.

The free ends of the adhesive layer and the vicinities of the two interfaces appear as critical adhesive regions due to the high normal and shear strains. Therefore, the probable initiation of damage can be expected in these regions. The through-thickness distribution and levels of strain and stress components are also formed depending on the variation of particle volume fraction through the adhesive thickness. However, the effect of the power index becomes negligible, especially for a lower maximum volume fraction of particles. In order to determine the effects of the maximum volume fraction of particles $V_{max}$ limiting the number of particles in the local adhesive composition and the power index $n$ tailoring the variation of volume fraction of particles $V_p$ through the adhesive thickness the variations of stress and strain components were evaluated along the upper aluminum adherend and lower aluminum adherend–adhesive interfaces (adhesive sides) and the adhesive mid-line. A grading direction was designated from the particle-rich adhesive–lower adherend interface to the neat adhesive–upper adherend interface.

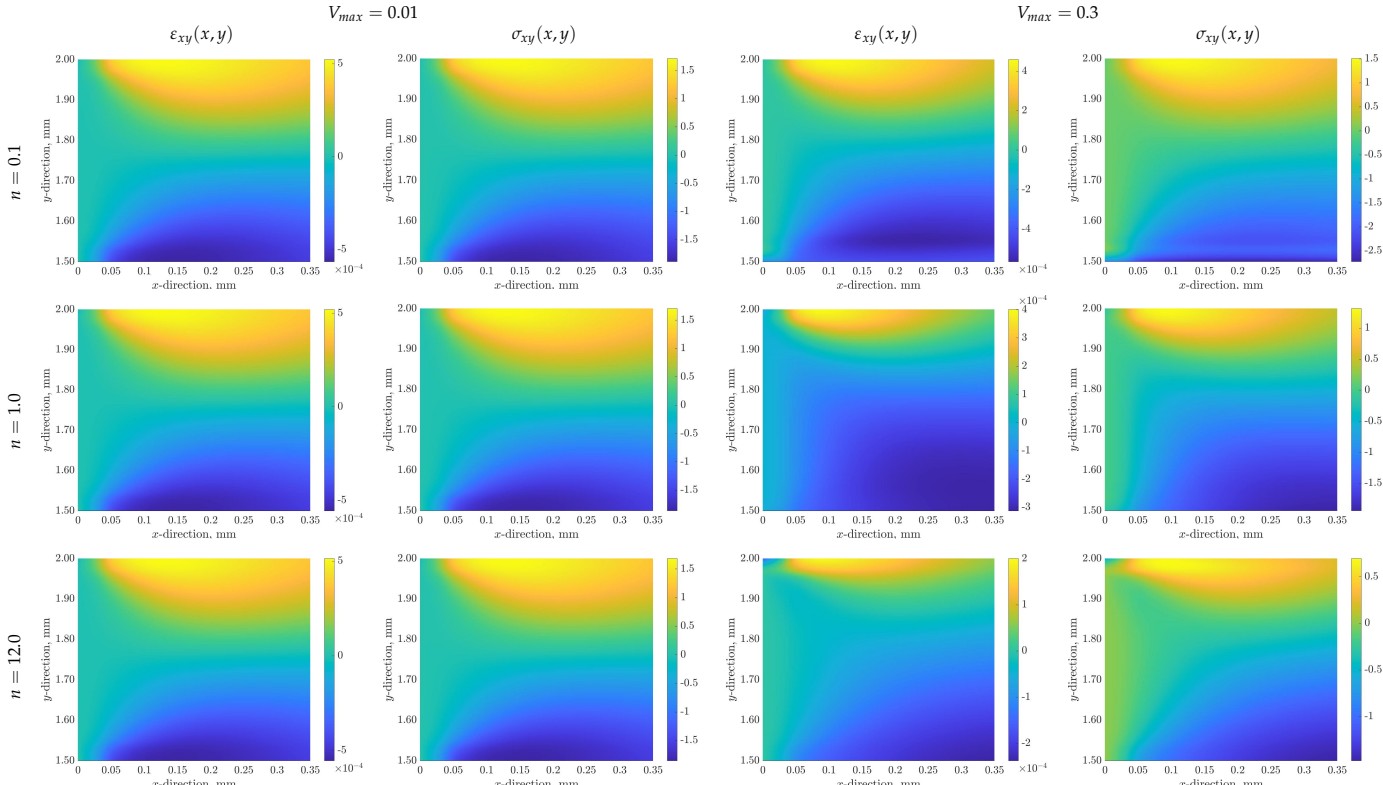

**Figure 9.** Effects of the gradient power index ($n$) and the maximum volume fraction of particles ($V_{max}$) on the shear strain $\varepsilon_{xy}(x, y)$ and stress $\sigma_{xy}(x, y)$ distributions around the left free-end of the PRA → NA functionally graded adhesive layer.

The normal strain $\varepsilon_{xx}$ increases uniformly from low levels in the center of the overlap region towards the free edges of the adhesive layer and becomes peak near the free edges (Figure 10). Similar variations appear along the lower and upper interfaces and the adhesive mid-line for all power index values. The normal strain levels are lower along both interfaces than those along the adhesive mid-line. The normal stress $\sigma_{xx}$ acts in compression and remains at high levels in a large overlap region and decreases towards the free edges of the adhesive layer, and then reaches zero levels (Figure 10). The effect of the power index becomes apparent on the normal strain and stress levels only for a high maximum volume fraction of particles, namely, the normal strain variations along the interfaces and adhesive mid-line are similar, but their levels decrease with increasing power index. The neat adhesive–upper adherend interface experiences higher normal stresses than the particle-rich adhesive–lower adherend interface. The power index affects normal stress levels rather than variation forms. The normal stress levels increase partly with increasing power index (enriched adhesive composition by particles) and decrease through the adhesive thickness from the neat adhesive–upper adherend interface to the particle-rich adhesive–lower adherend interface.

The normal strain $\varepsilon_{yy}$ exhibits almost high levels in a large overlap region and decreases towards the free edges of the adhesive layer and then reaches peak levels suddenly at the free edges (Figure 11). Even though a similar behavior is observed along both interfaces and adhesive mid-line, the variation form remains similar, but the normal strain levels along the interfaces are partly higher. For a higher maximum volume fraction of particles (0.3), enriching the adhesive composition by particles results in apparent changes in normal strain levels rather than their variation forms, especially along the adhesive mid-line and the particle-rich adhesive–lower adherend interface. The normal strain $\varepsilon_{yy}$ is more critical in a large overlap region in comparison to the adhesive-free edges. The normal stress $\sigma_{yy}$ acts in compression and exhibits similar variations along the lower and

upper interfaces and the adhesive mid-line, stays uniform at higher levels in a large overlap region and decreases uniformly towards the adhesive-free edges (Figure 11). The power index exhibits a negligible effect on the normal stress variations and levels for a low maximum volume fraction of particles. However, its influence becomes evident in the variation form and levels of normal stress, especially along the lower and upper interfaces, as the through-thickness adhesive composition is enriched by particles for a high maximum volume fraction of particles. A particle-rich adhesive composition variation results in lower normal strains but higher normal stresses.

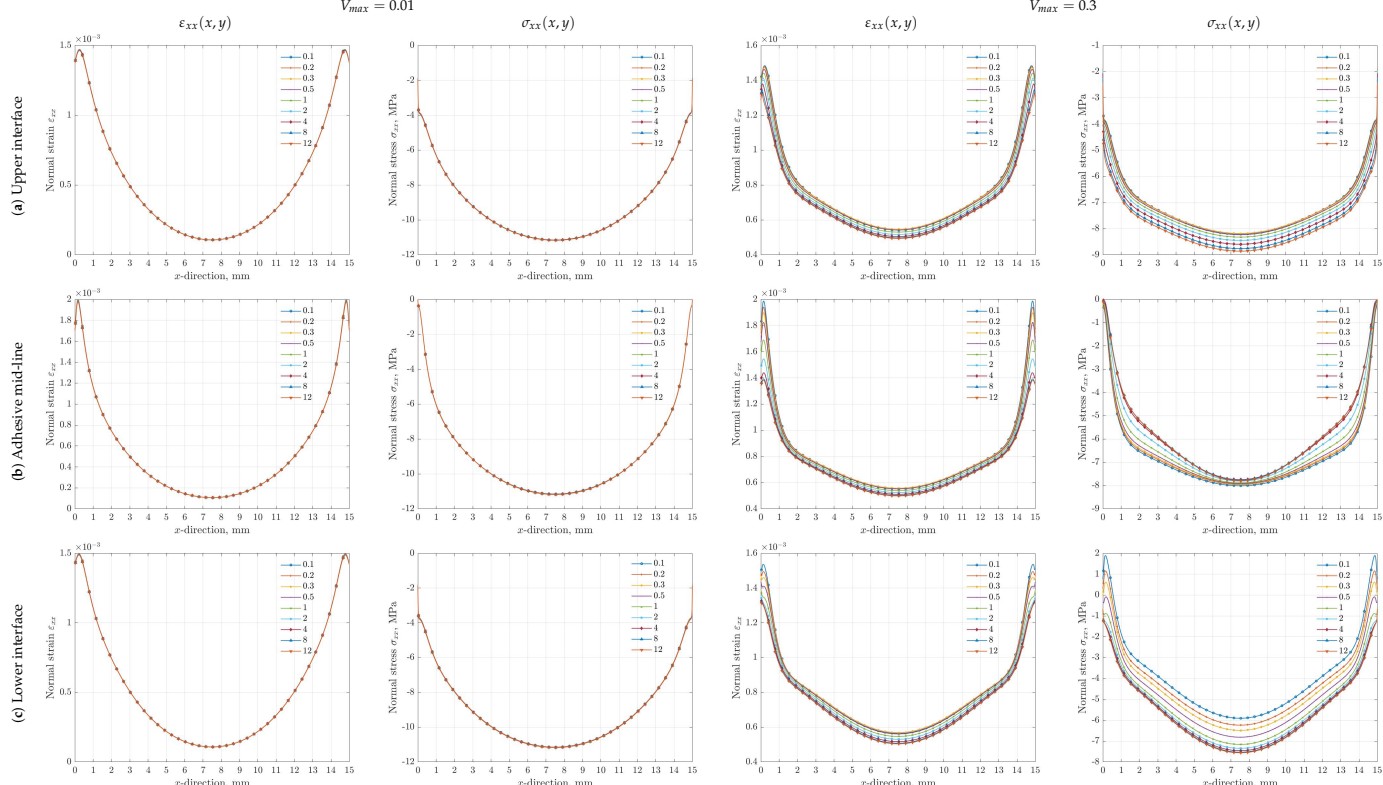

**Figure 10.** Effects of the gradient power index $(n)$ and the maximum volume fraction of particles $(V_{max})$ on the normal strain $\varepsilon_{xx}(x,y)$ and stress $\sigma_{xx}(x,y)$ variations along (**a**) the upper interface, (**b**) the mid-line and (**c**) the lower interface of the PRA → NA functionally graded adhesive layer.

The shear strain $\varepsilon_{xy}$ is uniform at negligible levels in a large overlap region and increases suddenly in the vicinity of the adhesive-free edges, and then decreases to zero levels at the adhesive-free edges (Figure 12). A symmetrical variation appears with respect to the center of the overlap region but in the opposite sense along both interfaces and adhesive mid-line. The lower and upper interfaces are of similar variations and levels but in the opposite sense. The shear strain exhibits a uniform increase along the adhesive mid-line in the center of the overlap region. In general, the power index and the maximum volume fraction of particles have a minor influence on the variations and levels of shear strain along both interfaces and adhesive mid-line. However, the peak values at the positions near the adhesive-free edges decrease with increasing power index. The shear stress $\sigma_{xy}$ has a symmetrical variation but in the opposite sense along both interfaces and adhesive mid-line with respect to the center of overlap region (Figure 12). It remains uniform at negligible levels in a large of overlap region and increases suddenly in the vicinity of the adhesive-free edges, and then becomes negligible at the adhesive-free edges. Both lower and upper interfaces experience similar variations and levels but in the opposite sense. The power index and the maximum volume fraction of particles have a negligible effect on the various forms and levels of shear stress. However, for a higher maximum volume fraction of particles (0.3), the peak shear stress levels at the positions near adhesive-free

edges decrease uniformly as the through-thickness adhesive composition is enriched by particles depending on the power index value.

**Figure 11.** Effects of the gradient power index ($n$) and the maximum volume fraction of particles ($V_{max}$) on the normal strain $\varepsilon_{yy}(x,y)$ and stress $\sigma_{yy}(x,y)$ variations along (**a**) the upper interface, (**b**) the mid-line and (**c**) the lower interface of the PRA $\rightarrow$ NA functionally graded adhesive layer.

The thermal strain and stress states occurring in an adhesive composition with a low volume fraction of particles are influenced negligibly by the through-thickness grading manner under a uniform temperature field. However, the free edges of the adhesive layer experience high axial and shear strain and stress levels, whereas the transverse strain and stress become apparent in a large overlap region. As the adhesive composition is enriched by particles at high volume fraction, the power index becomes more influential on the strain and stress levels rather than their variation forms.

Figure 13 shows the effects of power index, the maximum volume fraction of particle, and grading direction on the through-thickness variations and levels of strain components $\varepsilon_{ij}$ at the left free edge of adhesive layer ($x = 0$). The normal strain $\varepsilon_{xx}$ remains uniform at high levels in the vicinity of the adhesive mid-line, decreases uniformly towards both interfaces, and becomes minimum at both interfaces. An almost symmetrical through-thickness variation appears with respect to the adhesive mid-line. The power index influences only the normal strain levels rather than the variation manner, especially in a large region around the adhesive mid-line. As the through-thickness adhesive composition is enriched, the high normal strain levels reduce partly (increasing power index). In the case of an opposite grading direction, the general behavior of normal strain is the same as that in the previous grading direction. As the adhesive composition is enriched by particles at a higher volume fraction, the power index becomes more influential on both the strain variation and levels. Thus, the normal strain levels are uniform and maximum in a large region near the adhesive mid-line and become minimum at two interfaces. The through-thickness variation of normal strain is formed by the power index value. As the adhesive composition is enriched by particles, the normal strain levels decrease and become more uniform. The grading direction has a negligible effect on the variation form and levels of the normal

strain, and only turns down the normal strain variations between two interfaces. The normal strain $\varepsilon_{yy}$ is at uniform low levels in the adhesive region near the adhesive mid-line, increases uniformly from very near positions to both interfaces, becomes peak, and then decreases suddenly. However, its through-thickness variation is not symmetrical with respect to the adhesive mid-line on the contrary to that of the normal strain $\varepsilon_{xx}$, namely the adhesive zones near the neat adhesive–upper adherend interface experience higher normal strain levels. The power index has an effect on the normal strain levels rather than the through-thickness variation form for a lower maximum volume fraction of particles. An opposite grading direction does not influence the general behavior of normal strain, it turns down only the normal strain variations between two interfaces. However, both the levels and various forms of the normal strain change apparently depending on the power index value for a higher maximum volume fraction of particles (0.3). The shear strain $\varepsilon_{xy}$ becomes peak at both interfaces by increasing suddenly from uniform negligible levels in a large middle region through the adhesive thickness. The shear strain acts in the opposite sense in the adhesive regions near both interfaces. This response at the free ends of both interfaces arises due to the incompatible mechanical and thermal properties of bi-material interfaces. The variation and level of shear strain are not influenced by the power index for a low maximum volume fraction of particles, whereas the shear strain levels in the adhesive zones near both interfaces increase slightly for particle-enriched adhesive compositions. In addition, the grading direction does not influence the through-thickness variation of shear strain at the adhesive-free edge.

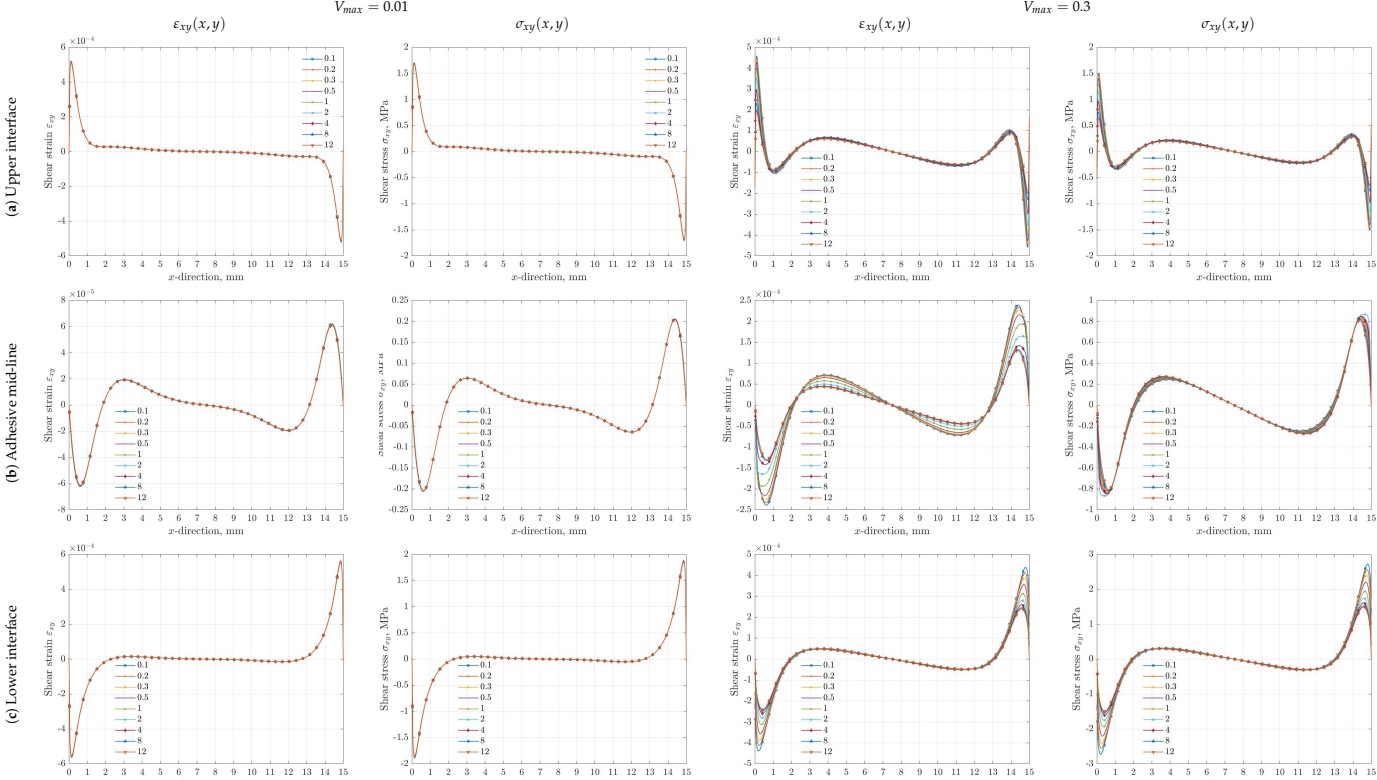

**Figure 12.** Effects of the gradient power index ($n$) and the maximum volume fraction of particles ($V_{max}$) on the shear strain $\varepsilon_{xy}(x, y)$ and stress $\sigma_{xy}(x, y)$ variations along (**a**) the upper interface, (**b**) the mid-line and (**c**) the lower interface of the PRA $\rightarrow$ NA functionally graded adhesive layer.

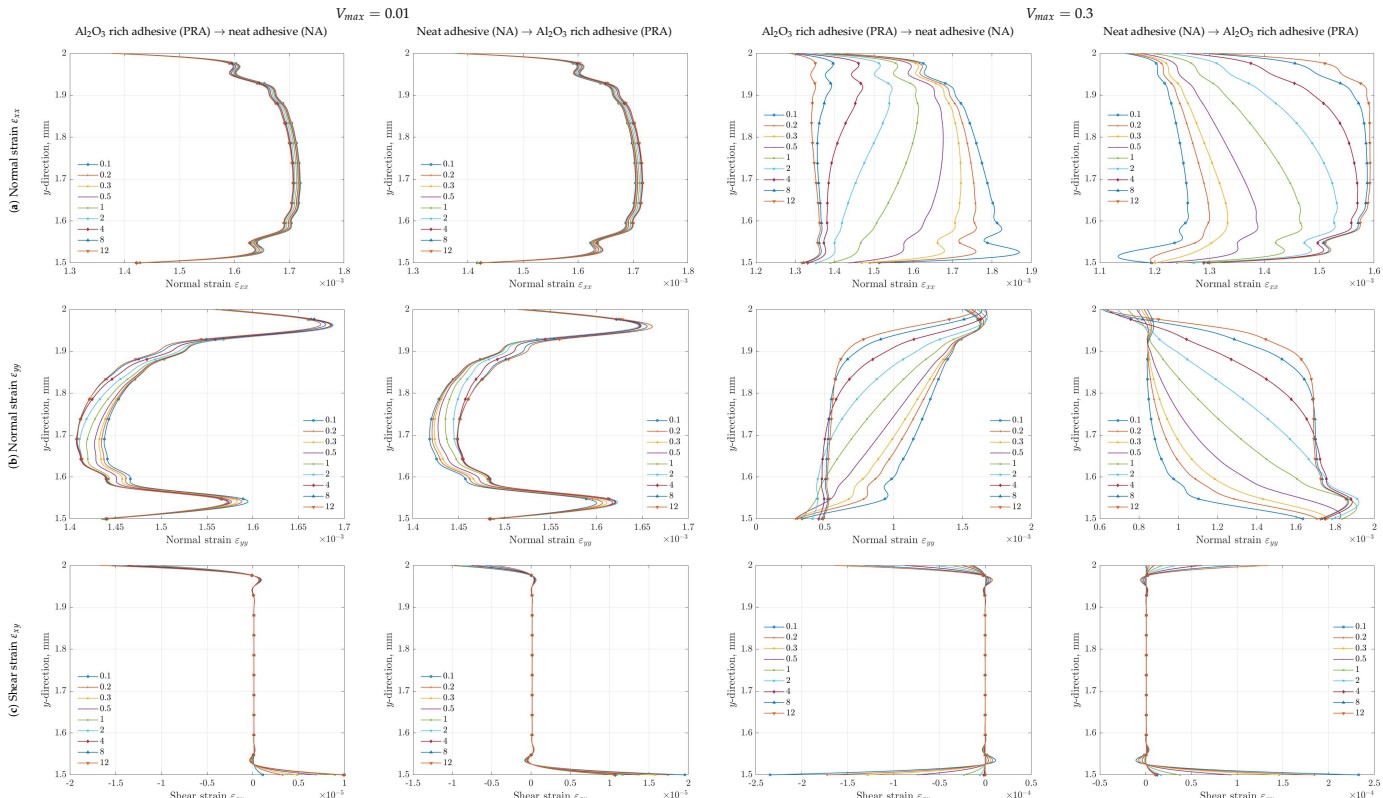

**Figure 13.** Effects of the gradient power index ($n$), the maximum volume fraction of particles ($V_{max}$) and the grading direction on (**a**) the normal strain $\varepsilon_{xx}$, (**b**) the normal strain $\varepsilon_{yy}$ and (**c**) the shear strain $\varepsilon_{xy}$ variations at the left free end ($x = 0$) of the functionally graded adhesive layer.

Figure 14 shows the effects of power index, the maximum volume fraction of particle, and grading direction on the through-thickness variations and levels of normal and shear stress components $\sigma_{ij}$ at the left free edge of adhesive layer ($x = 0$). The normal stress $\sigma_{xx}$ and shear stress $\sigma_{xy}$ are negligible at the free edge due to the zero-stress condition at the free edge. However, a sudden change near both interfaces appears due to the continuity conditions along the interfaces. The normal stress $\sigma_{yy}$ is uniform in the middle region of adhesive thickness and decreases towards a position near the interfaces, and then becomes maximum here by increasing towards both interfaces. The power index exhibits an evident influence on the through-thickness variation and levels of normal stress $\sigma_{yy}$ for only a high maximum volume fraction of particles (0.3). As the local adhesive composition is enriched by particles, the stress levels decrease and become more uniform. The adhesive zones near the particle-rich adhesive–lower adherend interface experience higher stresses. The grading direction can only upturn the through-thickness variations of stress components, whereas the general trend of stress variations remains the same.

As a result, the different mechanical and thermal properties of adherend and adhesive materials in a single lap joint result in thermal stresses in both adherends and adhesive layer under a uniform temperature field. The thermal conductivity of the adhesive layer can be improved by mixing Al$_2$O$_3$ particles, and the larger coefficient of thermal expansion of the adhesive layer, which is the main reason for incompatible-thermal strain, can be reduced by tailoring the adhesive composition. The adherend–adhesive interfaces exhibit sharp discontinuous thermal stresses. The discontinuous nature of thermal strains along bi-material interfaces can be smoothed by the power index. The free edges of the adhesive layer are critical due to the occurrence of high normal/shear strains and stresses. The gradient power index, which controls through-thickness volume fraction variation of particles, can influence the distribution and levels of strain and stress components only for a sufficiently high volume fraction of particles. The grading direction of the volume

fraction of particles in the adhesive layer is not influential because the temperature field is uniform, it can only reverse the low and high strain and stress regions because the neat adhesive–adherend interface and the particle-rich adhesive–adherend interface are relocated.

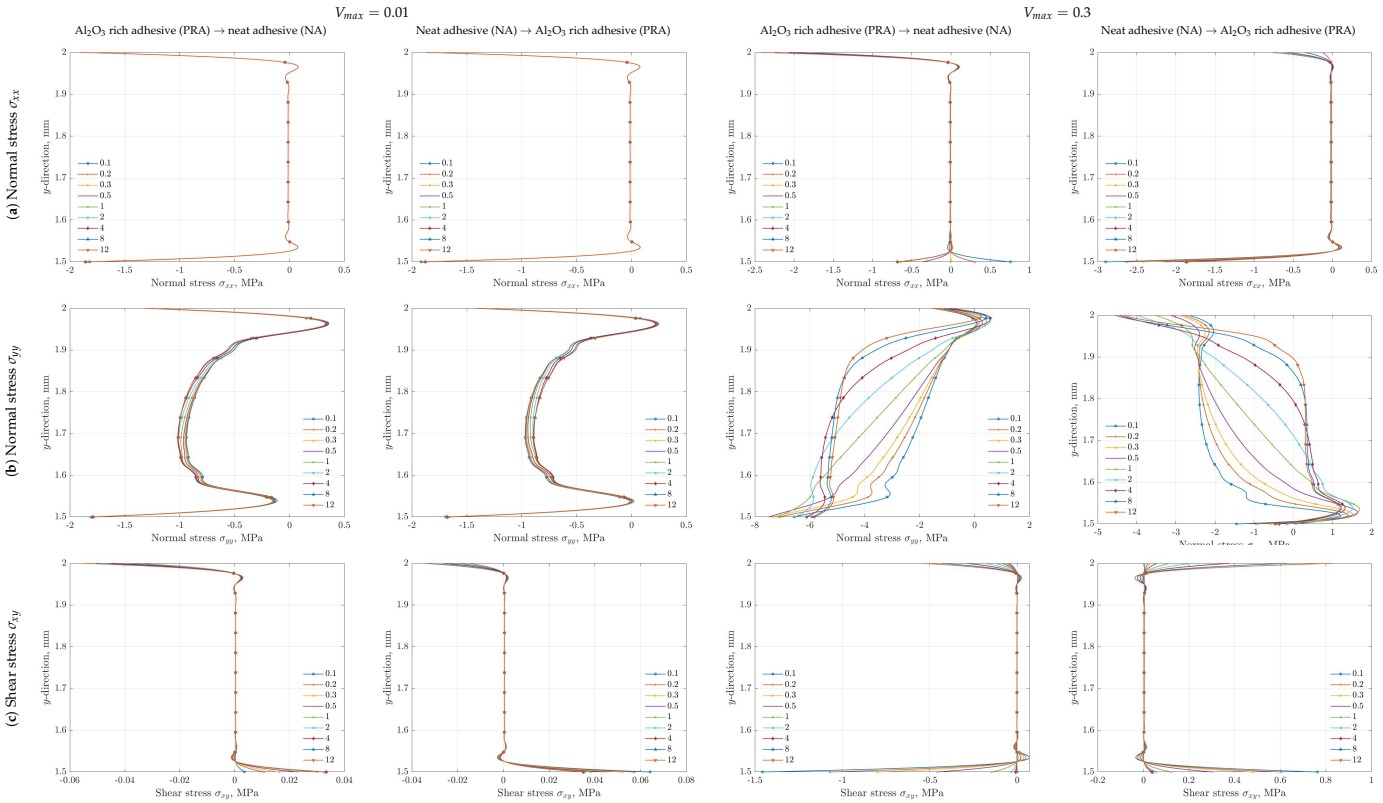

**Figure 14.** Effects of the gradient power index ($n$), the maximum volume fraction of particles ($V_{max}$) and the grading direction on (**a**) the normal stress $\sigma_{xx}$, (**b**) the normal stress $\sigma_{yy}$ and (**c**) the shear stress $\sigma_{xy}$ variations at the left free end ($x = 0$) of the functionally graded adhesive layer.

## 5. Conclusions

The thermal stress analyses of an aluminum single lap joint bonded with a through-thickness functionally graded adhesive layer subjected to a uniform temperature field show that:

- A uniform temperature field causes both adherends and adhesive layers to experience apparent deformation and thermal stress states due to the mismatches of thermal and mechanical properties of aluminum and neat or particle-reinforced adhesives.
- The normal ($xx$) and shear ($xy$) components of strain and stress remain uniform at very low levels in a large overlap region and reach peak levels around/at the free edges of $Al_2O_3$ reinforced adhesive layer, whereas a large of overlap region undergoes still high normal strain and stress components ($yy$).
- In order to control the deformation and stress states induced by a uniform temperature field, a functionally graded material concept was implemented. However, the influence of the gradient power index, which tailors the through-thickness variation of the volume fraction of $Al_2O_3$ particles, becomes apparent only for a sufficiently high volume fraction of particles.
- An excessive volume fraction of particles is not desired because the adhesion quality between adherend and particle-reinforced adhesive, namely, interfacial bonding strength, is deteriorated. In this respect, the volume fraction of particles is recommended to limit a reliable range of 0.01 and 0.1.

- The grading direction through the adhesive thickness between the neat adhesive–adherend interface and the particle-rich adhesive–adherend interface has a small influence on the variations of total strain and thermal stress components, and can partly affect only their levels because the temperature field is uniform. In the case of a nonuniform temperature field induced by a constant/variable applied heat flux, the effect of grading direction needs to be investigated for a large range of volume fractions of particles.

**Author Contributions:** Conceptualization, M.K.A. and J.N.R.; methodology, M.K.A. and J.N.R.; software, M.K.A.; validation, M.K.A. and J.N.R.; formal analysis, M.K.A.; investigation, M.K.A.; resources, M.K.A. and J.N.R.; data curation, M.K.A.; writing—original draft preparation, M.K.A.; writing—review and editing, M.K.A. and J.N.R.; visualization, M.K.A.; supervision, J.N.R.; project administration, M.K.A.; funding acquisition, M.K.A. All authors have read and agreed to the published version of the manuscript.

**Funding:** This research was funded by the Scientific Research Project Division of Erciyes University under the grant number [FUI-2019-9219].

**Data Availability Statement:** The data presented in this study are available on request from the corresponding author subjected to the permission requested from the funder, the Scientific Research Project Division of Erciyes University, on behalf of the Presidency of Erciyes University. The data are not publicly available.

**Conflicts of Interest:** The authors declare no conflict of interest. The funder had no role in the design of the study; in the collection, analysis, or interpretation of data; in the writing of the manuscript; or in the decision to publish the results.

## Abbreviations

The following abbreviations are used in this manuscript:

| | |
|---|---|
| $Al_2O_3$ | aluminum oxide |
| FGM | functionally graded material |
| NA | neat adhesive |
| PRA | $Al_2O_3$ particle-rich adhesive |
| *a*, *p* | adhesive, particle |
| *dx*, *dy* | increments between two neighbour grid points along the *x*- and *y*-directions |
| eps | specified error level |
| erru, errv | the differences of the calculated values of displacement components at iteration steps $k + 1$ and $k$ |
| *i*, *j* | indices of grid point along the *x*- and *y*-directions |
| *k* | iteration index |
| *n* | gradient power index |
| SumError | the total differences of the calculated values of displacement components at iteration step $k + 1$ and $k$ |
| $t_1$, $t_3$ | lower and upper adherend thicknesses |
| $t_2$ | adhesive thickness |
| $u_i$, *u*, *v* | displacement components along the *x*- and *y*-directions |
| $x_i$, *x*, *y* | spatial coordinate variables |
| $\bar{y}$ | the position relative to the lower adhesive interface |
| *E* | modulus of elasticity |
| *G*, $G_a$, $G_p$ | shear modulus (mixture, adhesive, particle) |
| *H* | joint height |
| *K*, $K_a$, $K_p$ | bulk modulus (mixture, adhesive, particle) |
| *L* | adherend and adhesive (joint) length |
| *T*, $T_0$, $T_{ref}$ | temperature |
| $\bar{T}$ | temperature difference |
| $V_a$, $V_p$ | volume fractions of adhesive and particles |
| $V_{max}$ | maximum volume fractions of particles |
| $\alpha$, $\alpha_a$, $\alpha_p$ | coefficient of thermal expansion (mixture, adhesive, particle) |

| | |
|---|---|
| $\psi(x, y)$ | a continuous, differentiable two-variable function |
| $\delta_{ij}$ | kronocker delta |
| $\lambda$, $\mu$ | Lamé's constants |
| $\nabla x$, $\nabla y$ | the central-difference operator of first-order partial derivative |
| $\nabla xx$, $\nabla yy$, $\nabla xy$ | the central-difference operator of second-order partial derivative |
| $\Gamma_x$, $\Gamma_y$ | the central-difference operator of first-order partial derivative |
| $\Gamma_{xx}$, $\Gamma_{yy}$ | the central-difference operator of second-order partial derivative |
| $\sigma_{ij}$ | stress components |
| $\varepsilon_{ij}$ | strain components |
| $\varepsilon_v$, $\varepsilon_{nn}$ | volumetric strain |
| $\nu$ | Poisson's ratio |
| $\rightarrow$, $\leftarrow$ | forward and backward sense |
| $-$, $+$ | lower and upper sides of adhesive interfaces |

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
