# Peer review of "Thermal Stress Formation in a Functionally Graded Al2O3-Adhesive Single Lap Joint Subjected to a Uniform Temperature Field"

_mca, doi:10.3390/mca28040082_

Round 1
Reviewer 1 Report
The study is interesting and the reviewer considers that this study can be published in this journal after some improvements:
- The geometry studied in not representative of a single lap joint. At the end of the each adherend there is the adherend that will affect the heating transfer at the end of bonded area.
The correct is consider as a bondline with a top and bottom plate.
- How was determined the maximum volume fraction of particles of 0.3. Why only was considered this value as maximum?
Author Response
Authors thank to Reviewer due to his/her constructive comments and recommendations. The possible explanations are made as follows:
a) a layered structure in a form of single lap joint is considered, its width along the z-direction is not taken into account. A constant temperature distribution is assumed through both adherends and adhesive layer. Consequently, a heat transfer model is not proposed and solved for this problem. In case any thermal boundary or internal conditions are specified the corresponding temperature distributions should be determined before the thermal stresses are calculated based on an uncoupled model. However, this is not the case for the present problem. In addition, we can not assume it as an individual very thin bondline since the adhesive thickness is not neglected, a functionally graded material composition is also defined, naturally, the theme of study comes from this point.
b) the maximum value of volume fractions of particles was not determined by any method. A larger value of 3.0 is thought to degenerate the adhesion quality between the adherend and adhesive zones relying on the experience obtained from the studies on the bonding mechanics of particle reinforced adhesives, composites. The present experimental studies in literature recommend this value to be kept smaller depending on the particle size as possible with a worry of degenerating adhesion quality as well as overall joint strength.
Reviewer 2 Report
The manuscript by the authors in involved with Navier equations of elasticity theory in conjunction with Mori-Tanaka’s homogenization approach for determining the strain and stress states in an aluminum single lap joint, which is bonded with a functionally graded Al2O3 micro particle reinforced adhesive layer subjected to a uniform temperature field. The work is interesting and is well prepared and presented. I just have some notes concerning clarity of the method and results, it would be very nice if a Nomenclature illustrating all the parameters appeared in the manuscript is added, for papers focusing on theory. Also, the results can be better presented if all the result figures (2,4-14) and font size can be enlarged a bit. Overall, it is a good and novel study on this subject, the text is clear and easy to read, the conclusions consistent with the evidence and arguments presented.
Author Response
Authors thank to Reviewer due to his/her constructive comments and recommendations. The following revisions are made as recommended:
a) a nomenclature illustrating all the key parameters appeared in the manuscript is included at the end of manuscript (line 664),
b) the size of all characters through Figures 2, 3, 4 to 14 is increased so that their readability can be improved as possible.